# Nur77 suppresses hepatocellular carcinoma via switching glucose metabolism toward gluconeogenesis through attenuating phosphoenolpyruvate carboxykinase sumoylation

Xue-li Bian[1,*], Hang-zi Chen[1,*], Peng-bo Yang[1,*], Ying-ping Li[1,*], Fen-na Zhang[1], Jia-yuan Zhang[1], Wei-jia Wang[1], Wen-xiu Zhao[2], Sheng Zhang[2], Qi-tao Chen[1], Yu Zheng[1], Xiao-yu Sun[1], Xiao-min Wang[2], Kun-Yi Chien[3] & Qiao Wu[1]

Gluconeogenesis, an essential metabolic process for hepatocytes, is downregulated in hepatocellular carcinoma (HCC). Here we show that the nuclear receptor Nur77 is a tumour suppressor for HCC that regulates gluconeogenesis. Low Nur77 expression in clinical HCC samples correlates with poor prognosis, and a Nur77 deficiency in mice promotes HCC development. Nur77 interacts with phosphoenolpyruvate carboxykinase (PEPCK1), the rate-limiting enzyme in gluconeogenesis, to increase gluconeogenesis and suppress glycolysis, resulting in ATP depletion and cell growth arrest. However, PEPCK1 becomes labile after sumoylation and is degraded via ubiquitination, which is augmented by the p300 acetylation of ubiquitin-conjugating enzyme 9 (Ubc9). Although Nur77 attenuates sumoylation and stabilizes PEPCK1 via impairing p300 activity and preventing the Ubc9-PEPCK1 interaction, Nur77 is silenced in HCC samples due to Snail-mediated DNA methylation of the Nur77 promoter. Our study reveals a unique mechanism to suppress HCC by switching from glycolysis to gluconeogenesis through Nur77 antagonism of PEPCK1 degradation.

[1] State Key Laboratory of Cellular Stress Biology, Innovation Center for Cell Signaling Network, School of Life Sciences, Xiamen University, Fujian Province, Xiamen 361102, China. [2] Department of Hepatobiliary Surgery, Zhong Shan Hospital, Xiamen University, Fujian Province, Xiamen 361005, China. [3] Molecular Medicine Research Center, Graduate Institute of Biomedical Sciences, College of Medicine, Chang Gung University, Tao-Yuan 333, Taiwan. * These authors contributed equally to this work. Correspondence and requests for materials should be addressed to Q.W. (email: qiaow@xmu.edu.cn).

Epidemiological surveys indicate that liver cancer is the second leading cause of cancer-related death, exceeded only by lung cancer. Hepatocellular carcinoma (HCC), which accounts for ~85 to 90% of primary liver cancers, affects more than 700,000 patients every year[1,2]. In addition to hepatitis B, hepatitis C and alcohol, metabolic syndromes, such as obesity and diabetes, have also been proposed as major risk factors for HCC[1]. Identification of new therapeutic targets for HCC is necessary, as no clear oncogene has been identified as responsible for HCC development to date, in contrast to other solid tumours, such as breast cancer, colon cancer and melanoma[3].

Metabolic reprogramming is considered one of the hallmarks of cancer[4]. In particular, cancer cells preferentially metabolize glucose to lactate, even with an ample supply of oxygen, in a process called aerobic glycolysis. Enhanced aerobic glycolysis provides selective advantages to cancer cells for cell proliferation, such as increased supply of metabolic intermediates essential for macromolecule biosynthesis. While substantial attention has focused on the regulatory roles of glycolysis in cancer cells, the role of gluconeogenesis, an inverse metabolic pathway to glycolysis, in cancer has also drawn attention. Because gluconeogenesis is a fundamental process in hepatocytes, it is important to understand whether and how gluconeogenesis influences HCC development.

Recently, the roles of gluconeogenic enzymes in anti-tumorigenesis were investigated. Fructose-1,6-bisphosphatase (FBP1), a rate-limiting enzyme in gluconeogenesis, was reported to be a tumour suppressor in renal and breast cancer. FBP1 inhibits renal carcinoma progression through antagonizing glycolytic flux and inhibiting the nuclear functions of HIF1 (ref. 5). FBP1 also suppresses PKM2 activation and increases the activity of mitochondrial complex I to hinder breast cancer cell proliferation[6]. Phosphoenolpyruvate carboxykinase (PEPCK) is another rate-limiting enzyme in gluconeogenesis. There are two isoforms of PEPCK, a cytoplasmic form (PEPCK1, PEPCK-C) and a mitochondrial isoform (PEPCK2, PEPCK-M), both of which catalyse the conversion of oxaloacetate to phospho-enolpyruvate. PEPCK1 is robustly expressed in the liver, kidney and brown and white adipose tissue[7] and is considered the main isoform involved in gluconeogenesis[8,9]. PEPCK1 function can be regulated by acetylation. p300-induced acetylation at Lys70, Lys71 and Lys594 affects the stability of human PEPCK1 and the impairment of gluconeogenesis[10]. In addition, the acetylation of yeast PEPCK1 at Lys19 and Lys514 is crucial for enzymatic activity and the ability of yeast cells to grow on non-fermentable carbon sources[11]. No other modification of PEPCK1 has been reported to date.

The nuclear receptor Nur77 (also named TR3), encoded by the immediate early gene Nr4a1, plays diverse and important roles in metabolic regulation. Hepatic expression of Nur77 elevates gluconeogenesis through the transcriptional upregulation of gluconeogenic genes, including G6pc, Fbp1 and Fbp2, and Eno3 (ref. 12). In contrast, Nur77 binding to the promoter regions of multiple genes involved in glucose metabolism in muscle promotes glucose utilization[13]. Mice with genetic ablation of Nur77 exhibit increased susceptibility to diet-induced obesity, as well as insulin and leptin resistance[14,15]. Nur77-targeting compounds can regulate the level of blood glucose in mice[16,17], and Nur77 is considered a promising therapeutic target for metabolic syndromes.

In addition to metabolic regulation, the inhibitory effects of Nur77 on colorectal cancer, melanoma and leukaemia have also been demonstrated[18–21]. Here we demonstrates that Nur77 suppresses HCC development by regulating glucose metabolism through an interaction with PEPCK1. However, PEPCK1 is sumoylated for degradation. Overexpression of Nur77 inhibits PEPCK1 sumoylation through competitively blocking Ubc9 targeting. Nur77 expression is suppressed in HCC samples due to the Snail-mediated DNA hypermethylation of the Nur77 promoter. This work demonstrates that both Nur77 and PEPCK1 are novel therapeutic targets for HCC.

## Results

**Nur77 is a suppressor for hepatocarcinogenesis.** Gene expression data from Oncomine demonstrates that Nur77 gene expression levels were substantially lower in HCC tissues than in normal liver tissues[22] (Supplementary Fig. 1a, left). With the development of HCC, Nur77 gene expression levels from dysplasia liver tissue to stage III of HCC gradually decreased to significantly lower levels[23] (Supplementary Fig. 1a, right). Consistent with these biostatistics, we also observed reduced gene and protein expression levels of Nur77 in clinical HCC samples and increased levels in the paired para-carcinoma samples (Fig. 1a). Of the 159 HCC samples, only 1.3% of tumour samples presented strong Nur77 expression, while 66.7% showed either low or a lack of Nur77 expression. In contrast, 30.8 and 17% of paired para-carcinoma samples displayed strong and low/no Nur77 expression, respectively. Importantly, Nur77 expression levels in clinical samples inversely correlated with the protein levels of the proliferative marker PCNA (Supplementary Fig. 1b). Nur77 expression decreased with HCC development from stage I to III (Fig. 1b), and patients with lower Nur77 levels were closely associated with poor clinical prognosis regardless of age and sex (Fig. 1c, Supplementary Table 1 and Supplementary Data 1). Together, these results implicate a repressive role for Nur77 in human HCC development.

Combined treatment of diethylnitrosamine (DEN) and carbon tetrachloride (CCl₄) can induce mouse hepatocarcinogenesis, which shares many features of the development of human HCC, including inflammation and fibrogenesis[24]. In this model, Nur77 knockout mice not only developed more severe hepatic fibrosis with elevated circulating IL-6 (Supplementary Fig. 1c,d) but also generated twice as many as tumour numbers and over four times larger tumours than WT mice (Fig. 1d). In addition, there was a higher weight ratio between liver and body in the knockout mice (Fig. 1d). A Nur77 deficiency significantly facilitated the proliferation of tumour cells in the analysis of Ki67 and PCNA in these tumour samples (Fig. 1e). Therefore, Nur77 inhibits hepatocarcinogenesis.

To further confirm this Nur77 inhibitory effect on HCC, we also utilized a streptozotocin (STZ) and high-fat diet (HFD)-induced hepatocarcinogenic mouse model, which mimics the nonalcoholic steatohepatitis (NASH)-hepatocarcinogenic process and closely follows human HCC progression[25]. In this model, Nur77 knockout promoted steatosis (Supplementary Fig. 1e), in accordance with a previous report[14,15]. Nur77 deficiency not only promoted tumour development but also accelerated the proliferation of tumour cells (Supplementary Fig. 1f,g). Serum insulin levels are elevated in Nur77-KO mice consuming an HFD[14], and the insulin/IGF pathway is involved in HCC development[26]. However, we observed no differences in insulin and IGF1 levels between WT and Nur77-KO mice in the STZ/HFD model (Supplementary Fig. 1h, left), which may be due to the toxicity of STZ to the insulin-producing β cells[27]. As similar results were obtained in a DEN/CCl₄ model (Supplementary Fig. 1h, right), it is likely that the Nur77 inhibition of HCC development does not rely on the insulin/IGF1 pathway.

The repressive role of Nur77 in liver cancer cell lines was further investigated. As expected, Nur77 gene and protein expression levels were substantially lower in most liver cancer cell lines compared with the normal human hepatocyte L02 (Supplementary Fig. 1i). Nur77 overexpression hindered cell

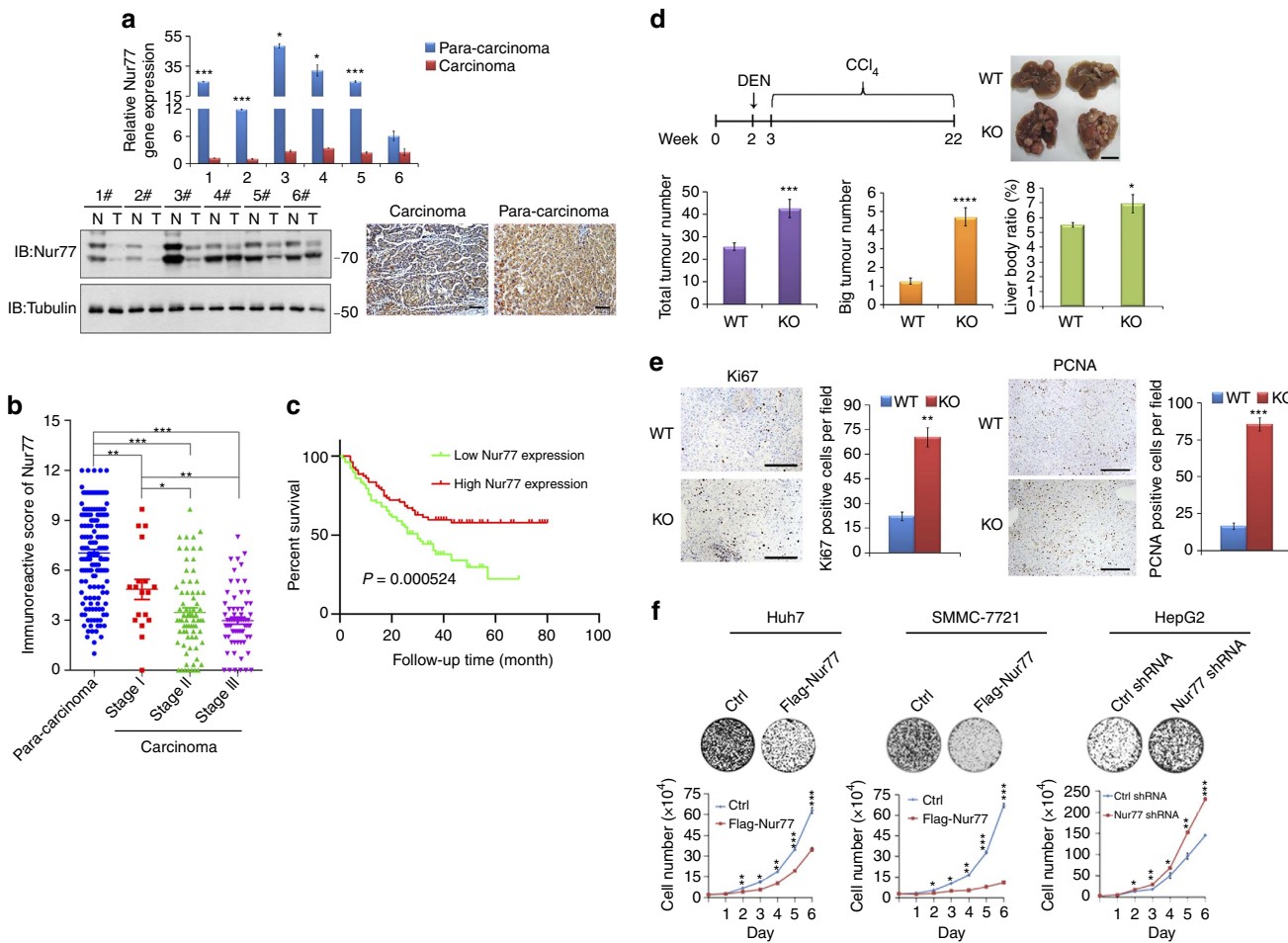

**Figure 1 | Nur77 is a tumour suppressor for hepatocellular carcinoma. (a)** The expression levels of *Nur77* gene (top) and protein (bottom) in clinical HCC (C) and paired para-carcinoma (P) samples detected by real-time PCR (top) or western blot and immunohistochemical staining (bottom). Scale bars, 100 μm. Tubulin was used to indicate the amount of loading proteins. **(b)** Scatter plot analysis of the immunoreactive score (IRS) of Nur77 in 159 para-carcinoma samples and 159 HCC samples grouped into stage I-III. **(c)** Kaplan-Meier survival curve shows the positive correlation between overall survival of HCC patients and Nur77 expression levels. Patients with Nur77 expression values below the 50th percentile were classified as lower Nur77 levels, while above the 50th percentile were classified as higher Nur77 levels. The median expression level was used as the cutoff. Survival information of 159 patients is available. **(d)** Top, the schematic overview of DEN/CCl₄ HCC mice model (left) and liver images (right). Bottom, tumour numbers and big tumour numbers (diameter > 5 mm), ratio of liver:body weight in DEN/CCl₄-induced HCC mice (*n* = 10). Scale bars, 1 cm. **(e)** Ki67 and PCNA expression levels in liver samples from DEN/CCL₄-induced HCC mice. Positive cells were quantified in 20 randomly selected fields per mouse (*n* = 6). Scale bars, 100 μm. **(f)** Nur77 inhibits proliferation of liver cancer cells determined by colony formation (top) and cell counting (bottom). Nur77 was stably expressed in Huh7 and SMMC-7721 cells or knocked down in HepG2 cells. Data were represented as means ± s.e.m. of at least three independent experiments. *$P < 0.05$; **$P < 0.01$; ***$P < 0.001$. The data were analysed using one-way ANOVA followed by Tukey *post hoc* test in (**a**,**b**,**f**) and two-tailed Student's *t*-test in (**d**,**e**).

proliferation and inhibited colony formation in Huh7 and SMMC-7721 cell lines that expressed relatively low levels of endogenous Nur77. In contrast, Nur77 knockdown in high-Nur77-expressing HepG2 cells significantly stimulated cell proliferation and colony formation (Fig. 1f and Supplementary Fig. 1j). However, Nur77 knockdown had a faint effect on L02 cell proliferation (Supplementary Fig. 1k), suggesting a selective Nur77 function on repression of HCC cell proliferation. Together, these data from clinical samples, mice, and cells support the notion that Nur77 is a suppressor of liver cancer.

It was reported that alterations of Wnt signalling and the p53 pathway are implicated in HCC development[3], and Nur77 can regulate cell proliferation via controlling Wnt signalling or the p53 pathway in colorectal cancer or osteosarcoma, respectively[19,28]. We thus evaluated whether these two signalling pathways were also involved in the current case.

Dominant-negative TCF4 (dnTCF4) has been shown to restrain Wnt signalling activity[29]. However, Nur77 overexpression efficiently inhibited Huh7 cell proliferation even in the presence of dnTCF4 (Supplementary Fig. 1l), precluding the involvement of Wnt signalling in the Nur77 inhibitory function on HCC. In addition, the fact that Nur77 efficiently inhibited cell proliferation in either HepG2 (p53-wildtype) or Hep3B (p53-null) cells (Fig. 1f and Supplementary Fig. 1m), infers that p53 is not required for Nur77 to inhibit HCC. Our data strongly suggest that Nur77 may retard HCC development through an unreported regulatory mechanism.

**Nur77 regulates the metabolic mode to inhibit HCC via PEPCK1.** Metabolic reprogramming is a hallmark of cancer[4]. Whether Nur77 influences HCC metabolism is unknown. To address

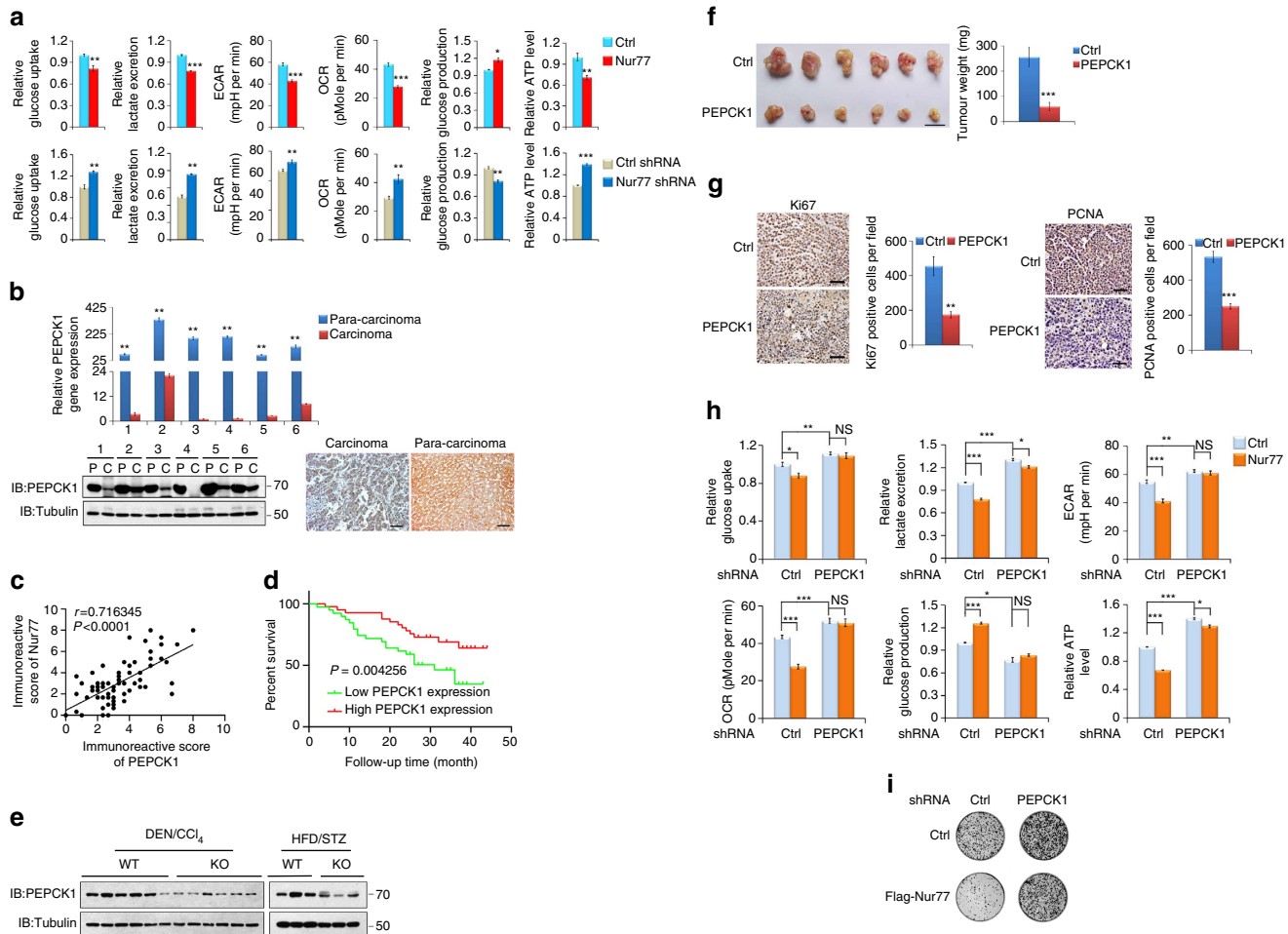

**Figure 2 | Nur77 regulates metabolism of hepatocarcinogenesis through PEPCK1.** (**a**) Analysis of glucose uptake and production, lactate excretion, ECAR, OCR and ATP level separately in Huh7 cells with Nur77 overexpression (top) or HepG2 cells with Nur77 knockdown (bottom). The ECAR bar values represent the glycolytic capacity, and the OCR bar values represent the ATP production-related oxygen consumption. The seahorse tracing curves of ECAR and OCR were shown in Supplementary Fig. 2a. (**b**) The expression levels of *Pepck1* gene (top) and protein (bottom) in clinical HCC (C) and paired para-carcinoma (P) samples were detected by real-time PCR (top) or western blot and immunohistochemical staining (bottom). Tubulin was used to indicate the amount of loading proteins. Scale bars, 100 μm. (**c**) The positive correlation of protein expression levels between Nur77 and PEPCK1 in HCC samples ($n = 82$). (**d**) Kaplan-Meier survival curve shows the positive correlation between overall survival of HCC patients and PEPCK1 expression levels. Survival information of 82 patients is available. (**e**) Endogenous PEPCK1 expression levels in HCC samples from DEN/CCl$_4$- and HFD/STZ-induced WT and Nur77-KO mice. (**f**) Images (left) and weight (right) of xenograft tumour in nude mice ($n = 6$). PEPCK1 was overexpressed in Huh7 cells that were then injected subcutaneously into the posterior flanks of nude mice. Scale bars, 1 cm. (**g**) The proliferative status of tumours is evaluated by Ki67 and PCNA immunohistochemical staining in xenograft tumour of nude mice. Positive cells were quantified in 20 randomly selected fields per mouse ($n = 6$). Scale bars, 25 μm. (**h,i**) PEPCK1 was knocked down first and then Nur77 was overexpressed in Huh7 cells. The glucose uptake and production, lactate excretion, ECAR, OCR and ATP level (**h**), as well as cell proliferation (**i**) were measured, respectively. Tubulin was used to indicate the amount of loading proteins. Data were represented as means ± s.e.m. of at least three independent experiments or of mice in the number indicated in the parenthesis. *$P < 0.05$; **$P < 0.01$; ***$P < 0.001$. The data were analysed using two-tailed Student's *t*-test in (**a,f,g**), one-way ANOVA followed by Tukey *post hoc* test in (**b,h**) and Pearson's chi-squared test in (**c**).

this issue, the possible participation of Nur77 in glucose metabolism was investigated. Overexpression of Nur77 in Huh7 cells not only reduced glucose uptake and lactate excretion but also suppressed the extracellular acidification rate (ECAR) and oxygen consumption rate (OCR) (Fig. 2a, top, Supplementary Fig. 2a, top). Knockdown of Nur77 in HepG2 cells reversed these trends (Fig. 2a, bottom and Supplementary Fig. 2a, bottom), suggesting that Nur77 could impede glucose catabolism. However, Nur77 overexpression significantly increased and Nur77 knockdown decreased glucose production (Fig. 2a), implying that gluconeogenesis, an ATP-consuming process of glucose anabolism, could be enhanced by Nur77. Because ATP was depleted upon Nur77 overexpression, yet accumulated

upon Nur77 knockdown (Fig. 2a), Nur77 might regulate glucose metabolism to inhibit HCC cell proliferation. Again, inhibition of gluconeogenesis by p53 may lead to tumour suppression[30,31]; however, Nur77 still regulated glucose metabolism in p53-null Hep3B cells (Supplementary Fig. 2b). Hence, p53 is indeed not involved in the Nur77-mediated glucose regulatory pathway. The fact that Nur77 could modulate glucose metabolism (Supplementary Fig. 2c) but not cell proliferation (Supplementary Fig. 1k) in normal liver L02 cells further suggests a different regulatory manner for Nur77 between HCC and non-transformed liver cells.

In contrast to the nuclear localization of Nur77 in L02 cells, Nur77 was distributed mainly in the cytoplasm in different liver

cancer cell lines (Supplementary Fig. 2d), which may be associated with Nur77 loss of most transcriptional activity in HCC cells, as transfection of Nur77 did not induce the mRNA and protein levels of its downstream target E2F1 (ref. 32) in HepG2, Huh7 and SMMC-7721 cells (Supplementary Fig. 2e). In contrast, Nur77, but not the Nur77 2G mutant that lost its DNA binding ability due to 2 Cys to Gly mutations at its zinc finger, still significantly promoted E2F1 expression in L02 cells (Supplementary Fig. 2e). Therefore, Nur77 inhibition of HCC cell proliferation may be independent of its transcriptional activity. The argument for this Nur77 action was further supported by the fact that Nur77 2G still inhibited the growth of SMMC-7721 cells (Supplementary Fig. 2f). It is reasonable to suggest that cytoplasmic Nur77 impairs glucose metabolism through interactions with metabolism-related proteins that are also located in the cytoplasm.

Interestingly, analysis by mass spectrometry of immunoprecipitated samples in SMMC-7721 cells indicated that PEPCK1, a rate-limiting enzyme in gluconeogenesis that is located in the cytoplasm, could be a novel Nur77 binding protein (Supplementary Fig. 2g,h). The interaction between the two proteins was confirmed via transfection or endogenous co-immunoprecipitation assays (Supplementary Fig. 2i). Furthermore, the expression levels of the two genes and proteins were lower in clinical samples but higher in para-carcinoma tissues (Figs 1a and 2b), and their protein levels showed a clear positive correlation (Fig. 2c). The patients who expressed lower PEPCK1 were also closely associated with poor clinical prognosis (Fig. 2d). In DEN/CCl$_4$- or HFD/STZ-induced mouse HCC samples, the ablation of Nur77 was accompanied by lower PEPCK1 expression levels (Fig. 2e). However, PEPCK1 knockdown in Huh7 and SMMC-7721 cell lines did not impair Nur77 expression levels (Supplementary Fig. 2j), suggesting that Nur77 is an upstream factor for PEPCK1 function. Therefore, Nur77 may inhibit HCC cell growth through PEPCK1 mediation.

PEPCK1 showed a significantly inhibitory effect on Huh7 xenograft tumour growth (Fig. 2f), with a clear decrease of Ki67 and PCNA levels (Fig. 2g). Consistently, PEPCK1 over-expression in Huh7 cells (Supplementary Fig. 2k, left) led to elevated glucose production, suppressed glucose uptake, lactate excretion, ECAR and OCR, depletion of ATP (Supplementary Fig. 2l,m, top), and inhibition of colony formation and cell proliferation (Supplementary Fig. 2n). PEPCK1 knockdown reversed these trends (Supplementary Fig. 2l,m, bottom), indicating that Nur77 and PEPCK1 may be involved in the same signalling pathway to co-regulate HCC development. Overexpression of Nur77 or Nur77 2G no longer altered glucose metabolism, ATP levels (Fig. 2h and Supplementary Fig. 2o), or cell growth (Fig. 2i and Supplementary Fig. 2p) when PEPCK1 was concurrently knocked down (Supplementary Fig. 2k, right). Together, it can be concluded that the presence of PEPCK1 is a prerequisite for Nur77 to inhibit HCC through the regulation of metabolic pathways.

**Sumoylation of PEPCK1 affects its stability**. Acetylation has been reported to be a main post-transcriptional modification for PEPCK1 degradation[10]. However, in the current study, when cell lysates were incubated with a de-sumoylation inhibitor NEM, we observed two additional specimens (indicated by arrowheads) above the typical PEPCK1 band (indicated by asterisk) in the mouse HCC samples that were positive via the anti-SUMO1 antibody (Fig. 3a, left). In DEN-induced HCC but not normal liver tissue (Fig. 3a, right) and in clinical HCC samples but not para-carcinoma samples (Fig. 3b), these two specific bands could also be detected. Similarly, the endogenous sumoylation of

PEPCK1 in HepG2 cells could be detected only in the presence of NEM (Fig. 3c, top). Transfection with both SUMO1 and the corresponding E2 enzyme, Ubc9 (Fig. 3c, bottom), or with exogenous PEPCK1 together with SUMO1 and Ubc9 (Supplementary Fig. 3a), induced PEPCK1 sumoylation. In contrast, transfection of a defective Ubc9 mutant (Ubc9$^{C93S}$) or a defective SUMO mutant (SUMO$^{GGAA}$)[33] induced no sumoylation (Supplementary Fig. 3a,b). However, the mutation of three critical acetylation sites (PEPCK1$^{Ac-3KR}$) in PEPCK1 (ref. 10) did not affect sumoylation (Supplementary Fig. 3c). Therefore, PEPCK1 is indeed sumoylated.

Transfection of SUMO1 and Ubc9 reduced the endogenous PEPCK1 expression levels in Huh7, SMMC-7721 and HepG2 cell lines (Fig. 3d, top), while the inactive Ubc9$^{C93S}$ or SUMO$^{GGAA}$ mutants could not (Supplementary Fig. 3d). Conversely, treatment of cells with anacardic acid, a sumoylation inhibitor[34], stimulated PEPCK1 expression with or without exogenous SUMO1 and Ubc9 in these cell lines (Fig. 3d, middle and bottom). Consistently, cycloheximide (CHX)-based protein stability assays revealed that the stability of PEPCK1 was dramatically reduced only when SUMO1/Ubc9, but not UBC9$^{C93S}$ or SUMO$^{GGAA}$, were transfected (Fig. 3e). Transfection of SUMO1/Ubc9 also led to a substantial degradation of PEPCK1$^{Ac-3KR}$ (Supplementary Fig. 3e). Therefore, PEPCK1 sumoylation, as well as acetylation, decreased protein stability.

Lys124, Lys471 and Lys473, which are conserved in many species, were identified as possible sumoylation sites in PEPCK1 by mass spectrometry analysis (Supplementary Fig. 3f). The K124R mutation completely abolished the larger sumoylated molecule, while the smaller sumoylated molecule was slightly affected by the K471R or K473R mutation (Fig. 3f). The double-mutant PEPCK1$^{K471\&473R}$ had no smaller sumoylated protein (Fig. 3f). Thus, Lys124, Lys471 and Lys473 are sites of PEPCK1 sumoylation. Although there are three PEPCK1 sumoylation sites, sumoylation on Lys124 but not Lys471 or Lys473 was linked to PEPCK1 stability (Fig. 3g and Supplementary Fig. 3g). These results strongly suggest that Lys124 plays a dominant role in sumoylation-regulated PEPCK stability. Interestingly, such sumoylation-induced PEPCK1 degradation occurs through the ubiquitination pathway, as shown by MG132 but not ALLM treatment, led to PEPCK1 accumulation in the presence of SUMO1/Ubc9 (Fig. 3h, left), which was accompanied by increased endogenous ubiquitin targeting to PEPCK1 but not to PEPCK1$^{K124R}$ (Fig. 3h, middle). For comparison, increased amounts of ubiquitin could still bind to PEPCK1$^{K471\&473R}$ (Fig. 3h, right), which is another indication that Lys124 is the only critical site for PEPCK1 stability.

**p300 enhances PEPCK1 sumoylation through acetylating Ubc9**. How PEPCK1 sumoylation is regulated is unknown. p300 was documented to induce PEPCK1 acetylation as an acetyltransferase[10]. Unexpectedly, p300 also dramatically enhanced SUMO1/Ubc9-induced or endogenous PEPCK1 sumoylation (Fig. 4a,b), thereby suppressing endogenous PEPCK1 expression levels in different liver cancer cells (Fig. 4b). Conversely, p300 knockdown effectively abolished Ubc9/SUMO1-induced PEPCK1 sumoylation (Supplementary Fig. 4a, left), thus enhancing PEPCK1 expression in different liver cancer cells (Supplementary Fig. 4a, right). However, without Ubc9 transfection, p300 alone could not significantly strengthen PEPCK1 sumoylation, even in the presence of SUMO1 (Fig. 4a, top, lane 4), which suggests that p300 is not directly responsible for PEPCK1 sumoylation. The fact that p300 could enhance either sumoylation of PEPCK1$^{Ac-3KR}$ (Supplementary Fig. 4b) or acetylation of PEPCK1$^{K124R}$ (Supplementary Fig. 4c)

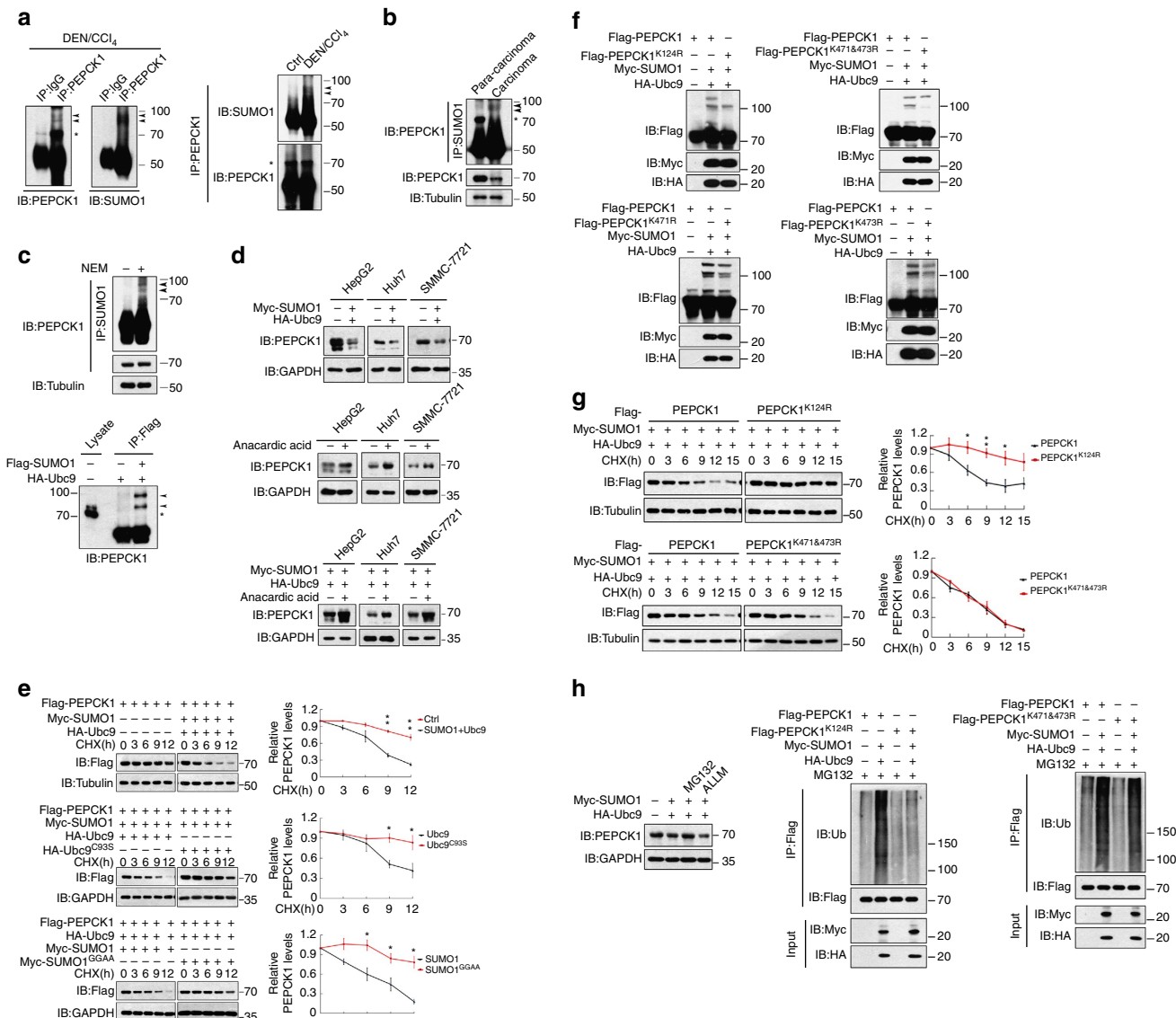

**Figure 3 | Sumoylation induces PEPCK1 degradation. (a)** In DEN/CCl$_4$-induced HCC samples (left) or normal livers (Ctrl) versus DEN/CCl$_4$-induced HCC samples (right), endogenous PEPCK1 was first immunoprecipitated and then incubated with anti-PEPCK1 antibody or anti-SUMO1 antobody, respectively. Asterisk represents position of PEPCK1 protein; arrowheads indicate sumoylation bands. **(b)** The levels of PEPCK1 and its sumoylation in clinical carcinoma and paired para-carcinoma samples. Endogenous SUMO1 was immunoprecipitated and then incubated with anti-PEPCK1 antibody. **(c)** Top, HepG2 cell lysates with or without NEM (20 mM) was immunoprecipitated with anti-SUMO1 antibody and then incubated with anti-PEPCK1 antibody. Bottom, SUMO1 and Ubc9 were transfected into 293T cells, and Flag-SUMO1 was immunoprecipitated and then detected by western blot with anti-PEPCK1 antibody. **(d)** Sumoylation effect on endogenous PEPCK1 expression in different liver cancer cell lines that were transfected with SUMO1/Ubc9 (top) or treated with sumoylation inhibitor, anacardic acid (20 μM, 12 h) with or without SUMO1/Ubc9 transfection (middle and bottom). **(e)** Sumoylation attenuated PEPCK1 stability in SMMC-7721 cells that were transfected with different plasmids and then treated with CHX (100 μg ml$^{-1}$) for different times (left). The amount of PEPCK1 protein was quantitated by software Image J (right). **(f)** Determination of the critical sites for PEPCK1 sumoylation in SMMC-7721 cells. The amount of PEPCK1 protein was normalized in each panel. **(g)** Lys124 site was important to stabilize PEPCK1 protein in SMMC-7721 cells with CHX (100 μg ml$^{-1}$) treatment. **(h)** Sumoylation-induced PEPCK1 degradation via ubiquitination pathway in SMMC-7721 cells that were treated with MG132 (5 μM) or ALLM (20 μM) for 12 h. The expression level of PEPCK1 (left) or ubiquitination levels of PEPCK1, PEPCK1$^{K124R}$ and PEPCK1$^{K471\&473R}$ (middle and right) were detected. NEM (20 mM) was added to cell lysates for repression of de-sumoylation in each sumoylation assay. Tubulin or GAPDH was used to indicate the amount of loading proteins. Data were represented as means ± s.e.m. of at least three independent experiments. *$P < 0.05$; **$P < 0.01$. The data were analysed using one-way ANOVA followed by Tukey *post hoc* test.

further demonstrated the dual functions of p300 in PEPCK1 sumoylation and acetylation. Furthermore, PEPCK1 stability, in the presence of Ubc9 and SUMO1, was distinct with or without p300 transfection (Supplementary Fig. 4d). It appears that p300 selectively strengthened the sumoylation of PEPCK1, but not other proteins, as transfection of p300 did not affect

global sumoylation induced by Ubc9 in different HCC cell lines (Supplementary Fig. 4e). These findings not only implicate a novel function of p300 in regulating the sumoylation of PEPCK1 but also suggest that acetylation and sumoylation of PEPCK1 are two dispensable p300 pathways that regulate PEPCK1 stability.

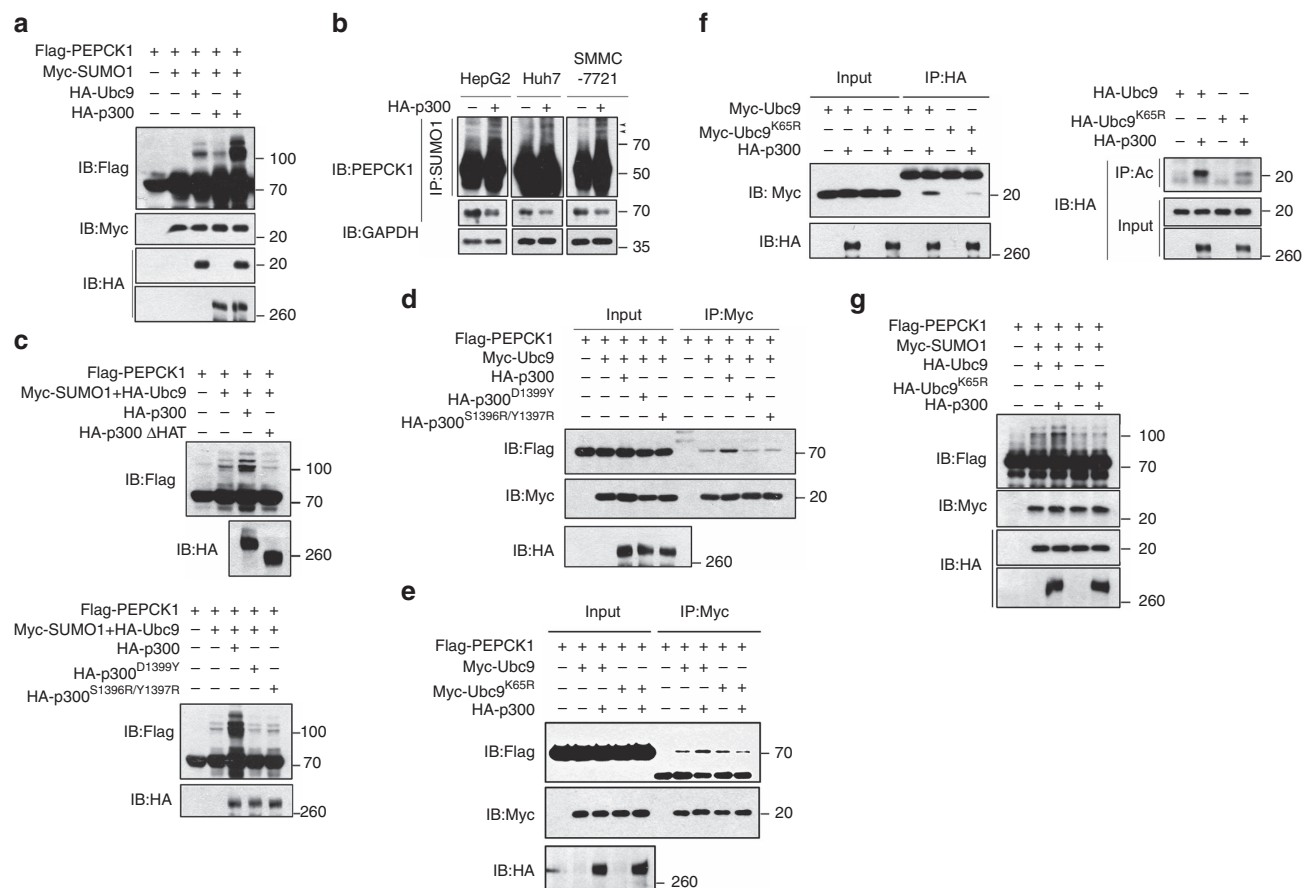

**Figure 4 | p300 enhances PEPCK1 sumoylation.** (**a**) p300-enhanced SUMO1/Ubc9-induced sumoylation in 293T cells. Different plasmids were transfected into cells, the sumoylation was indicated. (**b**) p300 inhibited endogenous SUMO1 and PEPCK1 levels in different liver cancer cell lines. p300 was transfected into cells and SUMO1 was immunoprecipetated and endogenous PEPCK1 sumoylation was indicated by its antibody. (**c**) p300-enhanced PEPCK1 sumoylation was dependent on its acetyltransferase activity. Different p300 inactive mutants were transfected into 293T cells as indicated. (**d**) p300 acetyltransferase activity was critical to enhance Ubc9 binding to PEPCK1 in 293T cells. (**e**) p300 could promote Ubc9, but not Ubc9$^{K65R}$, binding to PEPCK1 in 293T cells. (**f**) p300 interacted with Ubc9$^{K65R}$ less than with Ubc9 in 293T cells (left), and p300 could acetylate Ubc9 but not Ubc9$^{K65R}$, detected with anti-acetylation antibody (Ac, right). (**g**) Ubc9$^{K65R}$ had no effect on p300-enhanced PEPCK1 sumoylation in 293T cells. NEM (20 mM) was added to cell lysates for repression of de-sumoylation in each sumoylation assay.

The modes by which p300 participates in PEPCK1 sumoylation are of great interest. First, p300 activity is important for its enhanced sumoylation, as either the deletion of its histone acetyltransferase domain or the catalytically inactive mutations of p300$^{D1399Y}$ (ref. 13) and p300$^{S1396R/Y1397R}$ (ref. 10) resulted in no enhanced sumoylation of PEPCK1 (Fig. 4c), without impairing the interaction between the two proteins (Supplementary Fig. 4f). Second, only the native p300, but not these mutants, could enhance the interaction between PEPCK1 and Ubc9 (Fig. 4d). Third, although PEPCK1 interacted with both Ubc9 and its mutant Ubc9$^{K65R}$ with a mutation at its p300 acetylation site at Lys65 (ref. 35) (Supplementary Fig. 4g), p300 only facilitated Ubc9, but not Ubc9$^{K65R}$, targeting to PEPCK1 (Fig. 4e). p300 interacted better with Ubc9 (Fig. 4f, left) and only substantially acetylated Ubc9, but not Ubc9$^{K65R}$ (Fig. 4f, right). This situation consequently led p300 to enhance Ubc9-induced PEPCK1 sumoylation (Fig. 4g), supporting the notion that acetylation of Ubc9 modulates distinct SUMO targeting proteins[35]. Moreover, p300 still enhanced Ubc9 targeting to PEPCK1$^{Ac-3KR}$ (Supplementary Fig. 4h), thereby inducing sumoylation of PEPCK1$^{Ac-3KR}$ (Supplementary Fig. 4b). Combined with the fact that the *p300* and *Ubc9* gene and protein expression levels were higher in HCC samples than in

para-carcinoma samples (Supplementary Fig. 4i,j), it can be concluded that p300, through acetylating Ubc9 and facilitating acetylated Ubc9 binding to PEPCK1, could enhance PEPCK1 sumoylation, thus leading to lower PEPCK1 expression in HCC cells and clinical HCC samples.

**Nur77 inhibits HCC by attenuating PEPCK1 sumoylation.** Although Nur77 mildly enhanced transcriptions of gluconeogenic genes such as *Fbp2* and *Eno3* in several HCC cells (Supplementary Fig. 5a), it could not influence PEPCK1 mRNA levels (Supplementary Fig. 5b). Combined with the fact that Nur77 had a positive correlation with PEPCK1 protein level in clinical samples (Fig. 2c), we hypothesized that Nur77 may impair PEPCK1 expression through regulating its sumoylation. Indeed, Nur77 overexpression greatly attenuated SUMO1/Ubc9-induced sumoylation of PEPCK1 in different liver cancer cell lines, even in the presence of p300 (Fig. 5a). Increased expression of endogenous PEPCK1 was clearly detected in various cell lines (Fig. 5b) and over time after Nur77 transfection (Fig. 5c). This function of Nur77 on PEPCK1 expression was achieved by elevating the protein stability of PEPCK1, as stable Nur77 expression enhanced the half-life of PEPCK1 or PEPCK1$^{Ac-3KR}$ in the presence of

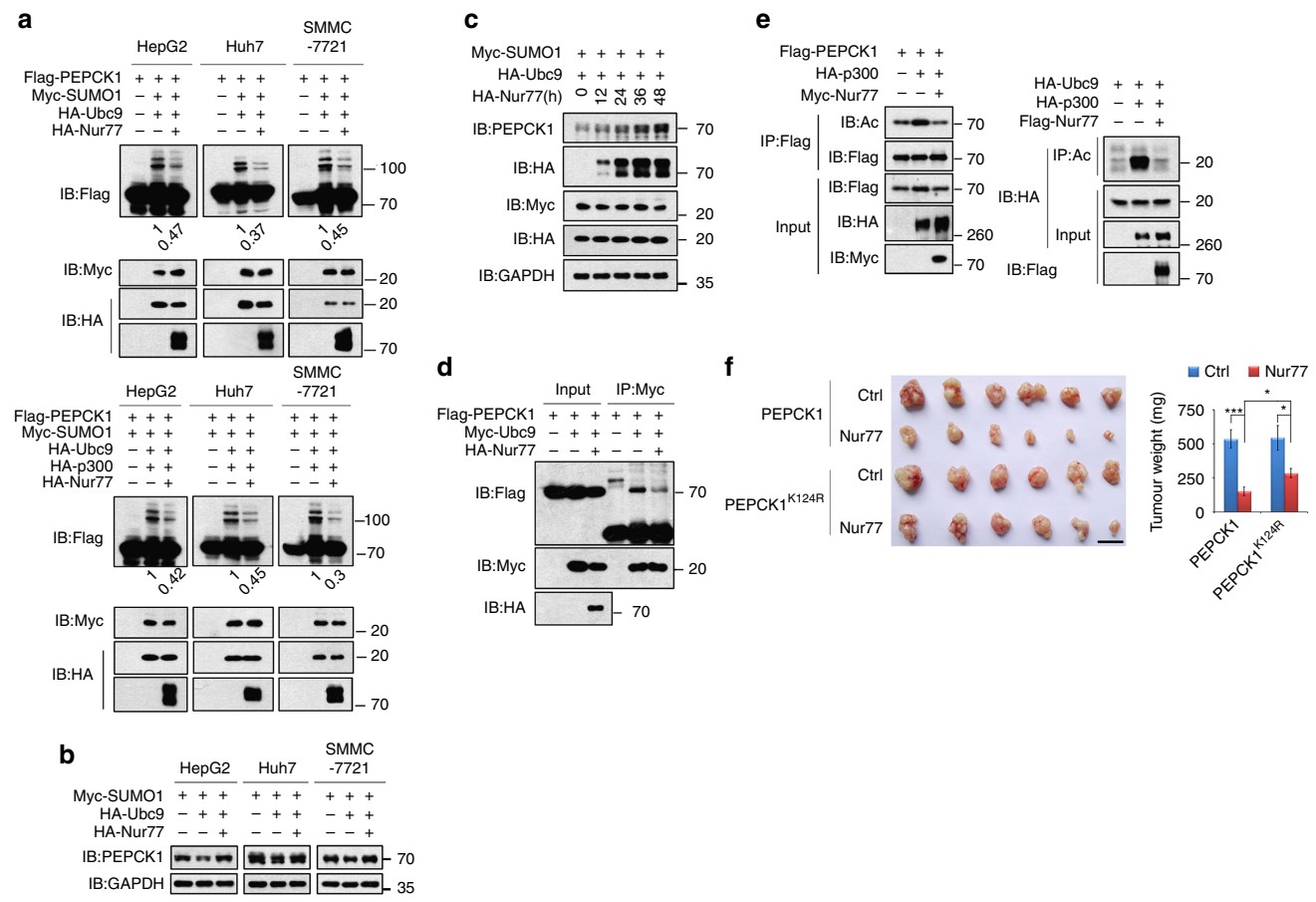

**Figure 5 | Activation of Nur77 attenuates PEPCK1 sumoylation and stabilizes PEPCK1. (a)** Nur77 abolished SUMO1/Ubc9-induced sumoylation (top), and p300-enhanced sumoylation (bottom) in different liver cancer cell lines. The amount of PEPCK1 sumoylation was quantitated by software Image J and presented under lanes of PEPCK1. **(b)** The role of Nur77 in elevating endogenous PEPCK1 expression levels in different liver cancer cell lines that were transfected with SUMO1 and Ubc9. **(c)** Nur77 elevated PEPCK1 protein expression in a time-dependent manner in HepG2 cells that were transfected with SUMO1 and Ubc9. **(d)** Nur77 blocked Ubc9 binding to PEPCK1 in 293T cells detected by co-IP assay. **(e)** Nur77 impaired p300 acetylation effect on PEPCK1 (left) or Ubc9 (right) detected with specific anti-acetylated lysine antibody (Ac) in 293T cells. **(f)** Images (left) and weight (right) of xenograft tumour in nude mice ($n = 6$). PEPCK1 or PEPCK1 K124R was stably overexpressed in Huh7 cells, and then Nur77 was further stably transfected into these cells. Cells were injected subcutaneously into the posterior flanks of nude mice. Scale bars, 1 cm. NEM (20 mM) was added to cell lysates for repression of de-sumoylation in each sumoylation assay. GAPDH was used to indicate the amount of loading proteins. Data were represented as means ± s.e.m. of mice in the number indicated in the parenthesis. $*P < 0.05$; $***P < 0.001$. The data were analysed using one-way ANOVA followed by Tukey *post hoc* test.

CHX (Supplementary Fig. 5c). Thus, Nur77 enhances PEPCK1 expression via modulating PEPCK1 sumoylation.

There are two possible mechanisms by which Nur77 downregulates PEPCK1 sumoylation and increases PEPCK1 expression: (1) Nur77 competes with Ubc9 for interaction with PEPCK1 and (2) Nur77 affects Ubc9 acetylation through impairment of p300 activity. Our data indicated that both mechanisms are involved. First, Nur77 and Ubc9 could competitively bind to the same 1–280 amino acid residue region in PEPCK1 that covers the critical sumoylation site Lys124 (Supplementary Fig. 5d), so that Nur77 disrupted the Ubc9-PEPCK1 interaction (Fig. 5d) and attenuated Ubc9-induced PEPCK1 sumoylation (Fig. 5a). Second, Nur77 overexpression abolished p300 activity to induce PEPCK1 or Ubc9 acetylation (Fig. 5e), which was accompanied by the impairment of p300-Ubc9 or p300-PEPCK1 association by Nur77 (Supplementary Fig. 5e,f), in accordance with our previous finding that Nur77 inhibited the p300 acetylation activity through their interaction[36].

To further evaluate the involvement of PEPCK1 sumoylation in Nur77-suppressed HCC samples, we performed xenograft tumour experiments in which PEPCK1 or PEPCK1[K124R] was first transfected into Huh7 cells, and then, Nur77 was stably expressed separately in these two cell lines (Supplementary Fig. 5g). As shown in Fig. 5f, transfection of PEPCK1 or PEPCK1[K124R] alone in the control group showed no obvious difference in tumour growth. However, further overexpression of Nur77 dramatically inhibited tumour growth by mediation of PEPCK1 but not PEPCK1[K124R]. This *in vivo* result consistently demonstrates that the repressive role of Nur77 on tumour growth occurs via PEPCK1 mediation that is associated with PEPCK1 sumoylation at Lys124.

**Snail represses *Nur77* gene expression in HCC.** DNA methylation has been known to play an important role in silencing gene expression. We found several CpG islands spanning the proximal regions of the Nur77 promoter (Supplementary Fig. 6a), suggesting that promoter methylation may be responsible for *Nur77* gene repression in HCC. To substantiate this possibility, SMMC-7721 and Huh7 cells were treated with 5-aza-2'-deoxycytidine (ADC), an inhibitor of DNA methyltransferase[37], to

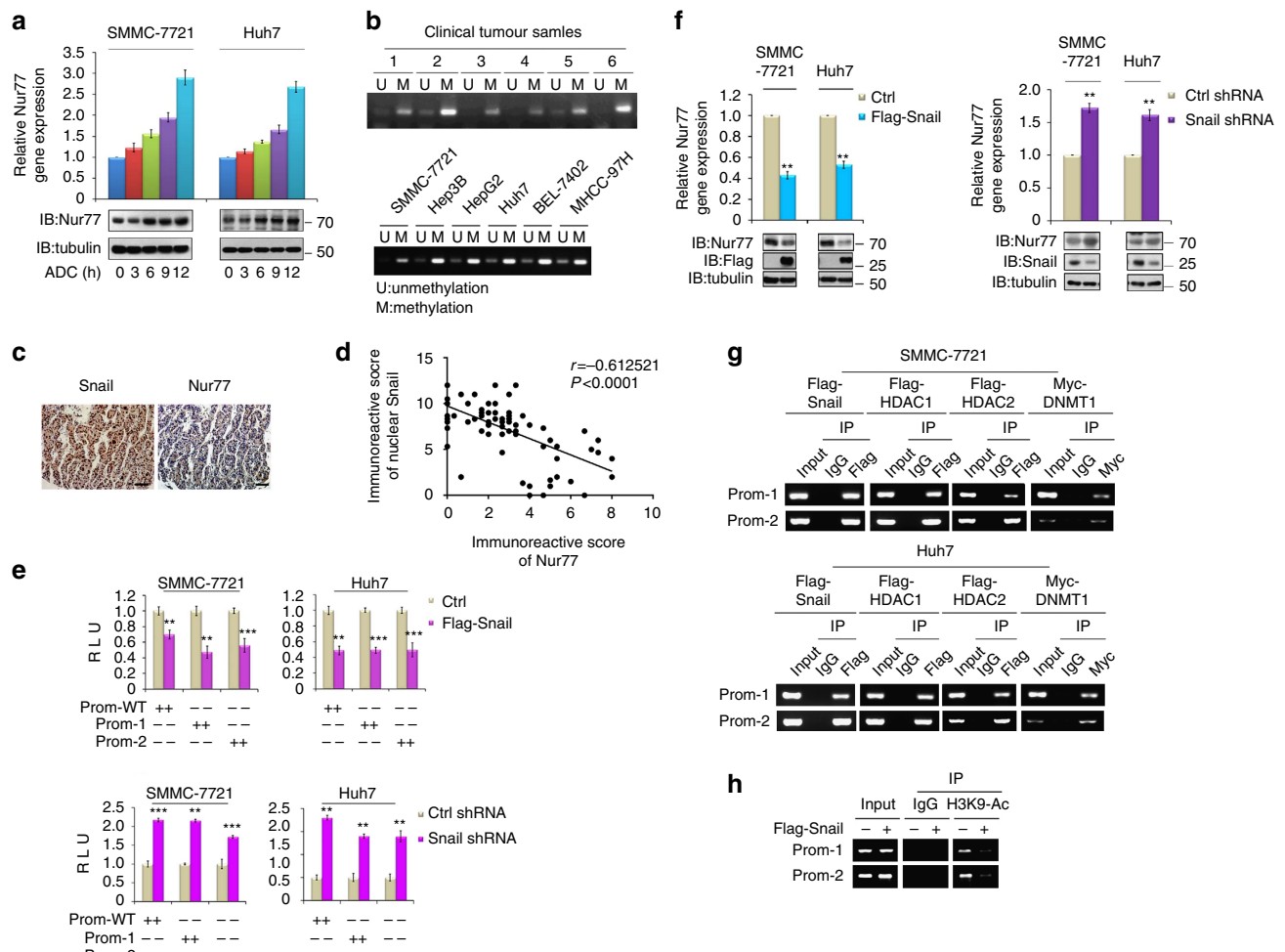

**Figure 6 | Snail downregulates *Nur77* gene expression via methylation pathway. (a)** The expression levels of Nur77 mRNA (top) and protein (bottom) in SMMC-7721 and Huh7 cells that were treated with ADC (10 μM) for indicated times. **(b)** Detection of methylation by methylation-specific PCR in clinical samples (top) and different liver cancer cell lines (bottom). U, unmethylation; M, methylation. **(c)** Snail and Nur77 expression levels detected in the same clinical HCC sample. Scale bars, 100 μm. **(d)** The negative correlation between the nuclear Snail and the protein levels of Nur77 in 82 clinical HCC samples. **(e)** Different reporters of Nur77 promoter, constructed as shown in Supplementary Fig. 6a, were transfected into Snail overexpression (top) or Snail knockdown (bottom) SMMC-7721 and Huh7 cell lines, and luciferase assays were perform to determine Nur77 promoter activity. **(f)** Overexpression of Snail (left) or knockdown of Snail (right) in SMMC-7721 and Huh7 cells affected expression levels of *Nur77* gene (top) and protein (bottom). **(g)** The recruitment of Snail, HDAC1, HDAC2 or DNMT1 to Nur77 promoter revealed by ChIP assays in SMMC-7721 and Huh7 cells. **(h)** Snail-mediated deacetylation of H3K9 on Nur77 promoter in Huh7 cells detected by ChIP assays. Tubulin was used to indicate the amount of loading proteins. Data were represented as means ± s.e.m. of at least three independent experiments. *$P < 0.05$; **$P < 0.01$; ***$P < 0.001$. The data were analysed using one-way ANOVA followed by Tukey *post hoc* test in (**e,f**) and Pearson's chi-squared test in (**d**).

restore the mRNA and protein levels of Nur77 (Fig. 6a). The methylation-specific PCR analysis further verified that Nur77 was frequently methylated in clinical HCC samples (Fig. 6b, top) and various liver cancer cell lines (Fig. 6b, bottom). Thus, promoter methylation may be associated with the repression of Nur77 in HCC development.

Snail is a suppressive transcription factor that binds to E-box DNA sequences to silence gene expression through the recruitment of chromatin remodellers and DNA methyltransferases[6,38,39]. Snail-mediated epigenetic regulation was recently reported to be involved in hepatocarcinogenesis[40]. After sequence analysis, we identified at least five Snail-binding E-boxes (CAGGTG) in the proximal region of the Nur77 promoter (Supplementary Fig. 6a), suggesting that Snail may be involved in the epigenetic regulation of Nur77 in HCC. Immunohistochemical analysis revealed that Snail was significantly enriched in the nuclei of HCC samples compared with their paired para-cancerous liver tissues (Supplementary Table 2), which in accordance with the importance of Snail nuclear localization for its functional activity[41]. More importantly, the nuclear expression level of Snail protein showed a significantly negative correlation with Nur77 expression in HCC samples (Fig. 6c,d). Therefore, nuclear Snail may act as a suppressive transcription factor to regulate the methylation of the Nur77 promoter. The Nur77 promoter was then cloned to generate several truncated promoter-luciferase constructs (Supplementary Fig. 6a). In both Huh7 and SMMC-7721 cells, transfection of Snail significantly suppressed the activity of Nur77 promoter, while Snail knockdown markedly increased the Nur77 promoter activity (Fig. 6e). As a result, Nur77 mRNA and protein levels were reduced by Snail overexpression, but increased by Snail knockdown (Fig. 6f). Further analysis showed that three E-boxes upstream of transcription start sites and one E-box localized between transcription start sites and start codon were

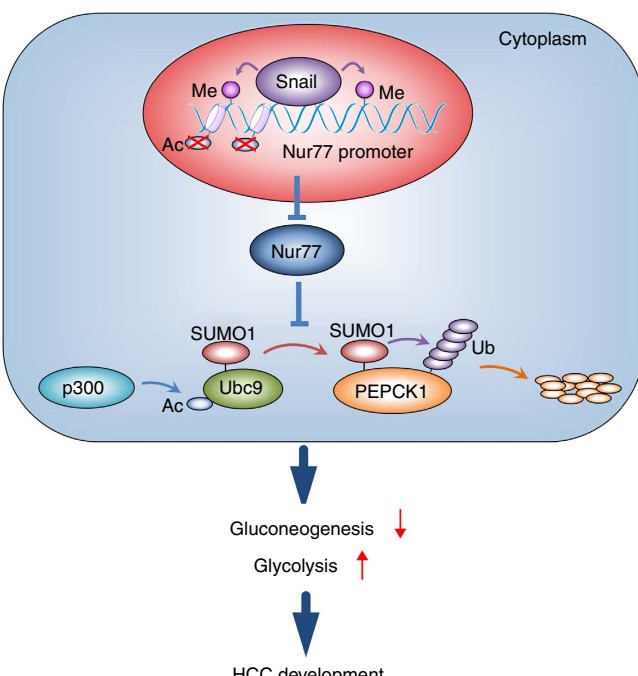

**Figure 7 | The functional model of Snail-Nur77-PEPCK1 pathway.** In HCC, PEPCK1 was sumoylated and p300 enhanced the sumoylation through acetylating Ubc9. Sumoylation led to PEPCK1 degradation via ubiquitination way, consequently boosted glycolysis and suppressed gluconeogenesis, promoted HCC development. Nur77 could stabilize PEPCK1 by competing Ubc9 binding to PEPCK1. Nevertheless, Snail with other factors suppressed expression of *Nur77* gene through DNA methylation and histone deacetylation on Nur77 promoter.

required for Snail to suppress Nur77 promoter activity, as mutations of these E-boxes (Mut-1 and Mut-3, but not Mut-2) abolished the suppressive effect of Snail on Nur77 promoter activity (Supplementary Fig. 6b). Therefore, Snail suppresses *Nur77* gene expression via binding to the E-boxes on the Nur77 promoter.

The recruitment of chromatin remodellers and DNA methyltransferase to silence the Nur77 promoter was further verified by ChIP assays, which clearly demonstrated that DNA (cytosine-5)-methyltransferase 1 (DNMT1) occupied the Nur77 promoter (Fig. 6g), in accordance with the report that the Snail-DNMT complex is responsible for DNA methylation[6]. The histone deacetylases HDAC1 and HDAC2 were also found to target the Nur77 promoter (Fig. 6g), suggesting histone deacetylation on the Nur77 promoter. Consistent with these results, H3K9 acetylation was dramatically decreased on the Nur77 promoter upon Snail overexpression (Fig. 6h). Together, it can be concluded that Snail-mediated H3K9 deacetylation and DNA methylation on the Nur77 promoter are critical for the repression of *Nur77* gene expression in HCC.

## Discussion

Although it has been reported that Nur77 is associated with apoptosis induction to suppress HCC[42], Nur77's role to impede HCC development has not been delineated. This study links Nur77 to HCC by altering glucose metabolism. Nur77 expression gradually decreased with HCC development, correlating with poor long-term prognosis. A deficiency of Nur77 in mice also significantly facilitated the occurrence of DEN/CCl$_4$- or HFD/STZ-induced HCC. This suppressive function of Nur77 is

associated with its interacting partner, the gluconeogenic enzyme PEPCK1. Overexpression of PEPCK1 in HCC upregulated gluconeogenesis and dampened glycolysis, leading to impaired ATP levels and arrested cell proliferation. However, PEPCK1 expression was suppressed after sumoylation, which facilitated the ubiquitination-proteasomal degradation of PEPCK1. PEPCK1 sumoylation could be augmented by p300-induced Ubc9 acetylation, but Nur77 competitively impeded Ubc9 targeting to PEPCK1 and impaired p300 acetylation activity to stabilize PEPCK1. In HCC development, Nur77 was silenced due to Snail-mediated DNA methylation and histone deacetylation on the Nur77 promoter (Fig. 7). This study not only describes a new Nur77 function in gluconeogenesis but also provides a unique strategy for HCC treatment through the activation of Nur77 to attenuate PEPCK1 sumoylation.

Gluconeogenesis is a process of energy consumption. In this pathway, three specific steps catalysed by PEPCK, FBP and glucose-6-phosphatase (G6Pase) are used to bypass the irreversible reactions of glycolysis with the cost of two GTP and four ATP molecules. However, only two molecules of ATP are generated in glycolysis[43]. In this regard, simultaneous activation of both pathways may result in a 'futile' cycling of glucose, which is detrimental to cell survival[44]. To avoid such futility, activation of either pathway should be exclusive to the other. For example, activation of FBP1, another gluconeogenic enzyme, will inevitably antagonize glycolytic flux[5,6]. Our results further demonstrated that the PEPCK1 activation of gluconeogenesis not only enhanced energy consumption but also attenuated the energy-producing process, that is, glycolysis and mitochondrial respiration, resulting in ATP depletion. In addition to the energy-producing functions, both aerobic glycolysis and mitochondrial respiration pathway also provide metabolic intermediates as cell building blocks[43,45]. Considering the essential functions of energy status and building blocks for cell doubling, it is not surprising that upregulation of gluconeogenesis by PEPCK1 finally results in suppression of HCC cell proliferation. As gluconeogenesis is dramatically impaired in malignant hepatocytes[46], it is possible that gluconeogenesis represents a metabolic barrier to retard HCC development.

PEPCK1 is one of the major regulatory factors in gluconeogenesis[8,9]. Although PEPCK1 promotes colon cancer and melanoma cell proliferation by increasing glucose and glutamine utilization toward anabolic metabolism[47,48], contradictory results indicate that PEPCK1 could inhibit cancer cell proliferation in human HCC and renal cell carcinoma cells[44,46,49]. These discrepancies in PEPCK1 function may be the result of employing different tissues and organs, as liver and kidney are the main organs for gluconeogenesis[9]. However, PEPCK1 could not promote glucose production in colon cancer or melanoma cells, where gluconeogenesis was not activated[50].

Liver-specific knockout of PEPCK1 in mice induces the accumulation of TCA cycle intermediates, which results in inhibition of the TCA cycle and oxygen consumption[51]. Similarly, PEPCK1 overexpression in skeletal muscle removes TCA cycle intermediates to ensure that the high rates of flux required to support exercise continues[52,53]. These results support the notion that PEPCK1 is a major cataplerotic enzyme that removes TCA cycle intermediates by converting oxaloacetate to phosphoenolpyruvate[54]. However, this function of PEPCK1 on TCA cycle flux may not hold true for cancer cells. Due to the high proliferative activity of cancer cells, TCA cycle intermediates are usually shifted for macromolecular synthesis, which is critical to supply sufficient building blocks for cell doubling[55]. Therefore, knockdown of PEPCK1 in cancer cells may not result in the accumulation of TCA cycle intermediates, which has also been reported by another paper[48]. In our study, PEPCK1 knockdown

in HCC cells leads to inhibition of glucose production but promotion of glucose utilization, which finally results in enhanced oxygen consumption in the TCA cycle.

In addition to acetylation[10], sumoylation of PEPCK1 was also responsible for PEPCK1 degradation. SUMO molecules could be ubiquitin antagonists, stabilizing target proteins by modifying the same lysine sites for ubiquitination. However, SUMO-modified proteins could also be targeted for ubiquitin-mediated degradation through specific ubiquitin E3 ligases[56], illustrating the opposite roles of sumoylation in protein stability. Sumoylation of PEPCK1 at Lys124 facilitates PEPCK1 degradation through the ubiquitination pathway. In conjunction with the fact that PEPCK1 is ubiquitinated and degraded by acetylation at Lys70, Lys71, and Lys594 (ref. 10), these two post-translational modifications of PEPCK1 appear mutually independent. Blocking acetylation did not influence the status of PEPCK1 sumoylation, and *vice versa*; however, these two modifications yielded the same biological outcome of degrading PEPCK1. It is also important that the functions of acetylation and sumoylation can be co-regulated by their upstream regulator p300, an acetyltransferase. p300 can either directly acetylate PEPCK1 or induce Ubc9 acetylation to enhance the interaction with PEPCK1, which then facilitates PEPCK1 sumoylation. In this regard, high expression of p300 (refs 57,58) as well as Ubc9 (ref. 59) in clinical HCC samples may partially explain the phenotype of low PEPCK1 expression through enhanced sumoylation modification.

Nur77 is not only a transcription factor for regulating gene expression but also participates in protein–protein interactions to exert its functions. The regulatory function of Nur77 on PEPCK1 may be manifested at least in three aspects. First, the Nur77 interaction with PEPCK1 competitively inhibited Ubc9 binding to PEPCK1 to impede PEPCK1 sumoylation and degradation. Second, Nur77 influenced PEPCK1 sumoylation via inhibiting p300 activity to acetylate Ubc9, which is in accordance with our previous finding that Nur77 can inhibit p300 activity through targeting the histone acetyltransferase domain of p300 (ref. 36). Third, Nur77 attenuated p300-induced PEPCK1 acetylation, which is also important for PEPCK1 stability[10]. These steps coordinately stabilized PEPCK1 to activate gluconeogenesis and then inhibit HCC cell proliferation.

This study also demonstrated that epigenetic regulation was responsible for the silencing of Nur77 in HCC. In this process, Snail plays a critical role besides its role in EMT[39]. In HCC, high Snail expression is associated with poor prognosis[60,61], opposing the function of Nur77. The role of nuclear Snail accumulation in HCC development is in accordance with its role as a repressive transcriptional factor to initiate DNA methylation and histone deacetylation of Nur77 promoter, which then down-regulate *Nur77* gene expression. This repressive function of Snail on Nur77 expression suggests that Nur77 is a link between Snail and gluconeogenesis.

## Methods

**Cell culture and transfection.** Human embryonic kidney (HEK293T) cell and human hepatoma cell lines (SMMC-7721, HepG2 and Hep3B) were purchased from the American Type Culture Collection (Manassas, VA, USA). Human hepatoma cell lines (Huh7, BEL-7402 and MHCC-97H) were purchased form Cell Bank in Chinese Academy of Sciences in Shanghai. Human liver cell (L02) was purchased from Cell Bank in Chinese Academy of Sciences in Kunming. Cells were cultured in DMEM Medium (for 293T, HepG2, Hep3B, Huh7, BEL-7402 and MHCC-97H cell lines) or RPMI-1640 medium (for SMMC-7721 and L02 cell lines), supplemented with FBS (10% (v/v), Gemini), penicillin (100 U) and streptomycin (100 μg ml$^{-1}$) (all are from BIO Basic Inc, Shanghai, China). The cell lines were routinely tested and found negative for mycoplasma. Transfection was performed using TurboFect Transfection Reagent (ThermoFisher Scientific, Bremen, Germany) for all of cell lines, besides 293T cells that were transfected using the calcium phosphate method. Cells were used 16–24 h after transfection.

**Antibodies and reagents.** Anti-HA (cat. #H-9658; 1:5,000 dilution for western blot), anti-Flag (cat. #F-1804; 1:5,000 dilution for western blot), anti-PARP (cat. #P7605; 1:5,000 dilution for western blot) and anti-β-tubulin (cat. #T4026; 1:5,000 dilution for western blot) antibodies were purchased from Sigma. Anti-PCK1(cat. #D12F5; 1:2,000 dilution for western blot), anti-Nur77 (cat. #D63C5; 1:2,000 dilution for western blot), anti-SUMO1 (cat. # 4930S; 1:2,000 dilution for western blot), anti-acetylated lysine (cat. # 9681S; 1:2,000 dilution for western blot) and anti-snail (cat. #C15D3; 1:2,000 dilution for western blot) antibodies were from cell signalling technology. Anti-PEPCK-C (cat. #P-14; 1:2,000 dilution for western blot) and Anti-Ubiquitin (cat. #sc-8017; 1:2,000 dilution for western blot) antibodies were from Santa Cruz biotechnology (Santa Cruz, CA, USA). Anti-Myc (cat. # 11667149001; 1:5,000 dilution for western blot) antibody was from Roche. Anti-SUMO1 (cat. #332400; 1:2,000 dilution for western blot) antibody was from ThermoFisher scientific. Anti-GAPDH (cat. #M20006; 1:5,000 dilution for western blot) antibody was from Abmart biotechnology. Anti-acetylated lysine (cat. # 05–515; 1:2,000 dilution for western blot) antibody was from Millipore. Anti-PEPCK1 (cat. #ab133603) and Anti-Snail (cat. #ab180714) antibodies for immunohistochemistry were from Abcam. Anti-Nur77 (cat. #ARP31941T100) antibody for immunohistochemistry was from Aviva Systems Biology. anti-ki-67 (cat. # 12202S) antibody for immunohistochemistry was from cell signalling technology.

The chemical reagents N-Ethylmaleimide (cat. #128-53-0), and 5-Aza-2-deoxycytidine (cat. #2353-33-5) were purchased from Sigma-Aldrich (St Louis, MO, USA). Anacardic acid (cat. #16611-84-0) was from Abcam (Cambridge, UK). CHX (cat. #2112s) was from cell signalling technology (Beverly, Massachusetts, USA). MG132 (cat. #s2619) was from Selleckchem (Houston, TX, USA). 2-NBDG (cat. # N13195) was from Invitrogen (Gaithersburg, MD, USA).

**Clinical tumour samples.** Fresh HCC cancerous and para-cancerous tissues were obtained from Zhongshan Hospital, Xiamen University with patient informed consent and approval of the Medical Ethical Committee of Zhongshan Hospital. These samples were used for real-time PCR, western blot and MSP analysis. Liver cancer and their corresponding adjacent liver tissue microarrays were obtained from Shanghai Outdo Biotech Company, China. Nur77 expression was determined by immunohistochemical staining in 159 cases (Cat. No. HLiv-HCC180Sur-03, HLiv-HCC180Sur-04), and PEPCK1 and Snail expressions were determined in 82 cases (Cat. No. HLiv-HCC180Sur-03) with complete patient diagnosis and survival information.

**Animal models.** Wildtype (WT) and Nur77 knockout (KO) mice (C57BL/6J background) were purchased from the Jackson Laboratory in USA, and maintained in 12 h light/12 h dark cycles with free access to food and water. Tumour number or tumour weight was evaluated in a blinded manner between WT and KO mice but not random allocation. All of the animal experiments were approved by the Animal Ethics Committee of Xiamen University (acceptance no.: XMULAC20120030).

For DEN/CCl$_4$-induced hepatocarcinoma model[24], 15-day-old male mice were intraperitoneally injected with diethylnitrosamine (DEN, dissolved in PBS, 25 mg kg$^{-1}$ body weight). One week later, the mice were intraperitoneal injected with 10% carbon tetrachloride (CCl$_4$ in 0.5 ml corn oil per kg body weight) once a week for 22 weeks.

For HFD/STZ-induced hepatocarcinoma model[25], 2-day-old male mice were intraperitoneally injected with STZ (dissolved in citric acid buffer, 200 μg per mice). Four weeks later, mice were fed with high-fat diet (HFD, 60% of calories from fat, Research Diets, Inc.) until 20-week-old and then killed. The tumour numbers and liver weights were recorded and ratios of liver:body weight were calculated.

For nude mice model, male nude mice (BALB/c, 18–20 g, 6–7 weeks old) were obtained from Laboratory Animal Center of Xiamen University. PEPCK1 or PEPCK1 K124R was stable expressed in Huh7 cells, which were further transfected with or without Nur77. Cells (2 × 10$^6$ cells) were suspended in 150 μl PBS and injected subcutaneously into the either posterior flanks of mice. Twenty-four days after inoculation, mice were killed and the tumour weights were recorded.

**Immunohistochemical staining and scoring.** The formalin-fixed paraffin-embedded samples were deparaffinized with xylene and ethanol for further peroxidase immunohistochemistry staining using DAKO EnVision System (Dako Diagnostics, Zug, Switzerland). The slides were incubated with antibody diluted in Tris-buffered saline containing BSA for overnight at 4 °C. Peroxidase-labelled polymer and substrate-chromogen were then employed to visualize the staining of the interested protein.

Immunohistochemical staining was semiquantitated using the immunoreactive score (IRS) system[62]. IRS gives a range of 0–12 as a product of multiplication between proportion score (0–4) and staining intensity score (0–3). A proportion score represents the estimated percentage of positive-staining cells (0: no positive cells; 1: <10% of positive cells; 2: 10–50% positive cells; 3: 51–80% positive cells; 4: >80% positive cells). An intensity score represents the average intensity of staining (0: no staining; 1: yellow, 2: claybank; and 3: tawny). Each sample was evaluated by three persons in a blinded manner and the mean score was considered

as the final IRS. The IRS was interpreted as follows: 0–1: negative; 2–3: mild expression; 4–8: moderate expression; 9–12: strong expression.

**Plasmid constructions.** Human p300 construct with HA tag was established previously[63]. p300-ΔHAT was created by deleting amino acid residues 1413–1721 from p300. Human HA-Ubc9 and Myc-SUMO-1 were kindly provided by Olle Larsson (Karolinska Institute, Stockholm, Sweden). Flag-Snail is a gift from ChunDong Yu (Xiamen University, Fujian, China). The point mutations of Nur77, PEPCK1, p300, HA-Ubc9 or MYC-SUMO-1 were generated with the Quick Change mutagenesis kit (Stratagene, La Jolla, CA, USA) and verified by sequencing. Primer information for the various constructs is available upon request.

**Immunoprecipitation and western blot.** Immunoprecipitation was performed as described previously[64]. Briefly, cells were lysed on ice with lysis buffer ELB (150 mM NaCl, 100 mM NaF, 50 mM Tris-HCl (pH 7.6), 0.5% Nonidet P-40 (NP-40) and 1 mM PMSF), cell lysates were then incubated with the appropriate antibody for 2 h and subsequently with protein G-Sepharose beads for another 1 h. The protein-antibody complexes that were recovered on beads were subjected to western blot analysis after separating by sodium dodecyl sulfate-polyacrylamide gel electrophoresis (SDS-PAGE), and then transferred to polyvinylidene difluoride (PVDF) membranes (Millipore). The membrane was probed with primary antibodies, and then incubated with the secondary antibodies. The immunoreactive products were detected using enhanced chemiluminescence (Pierce, Socochim SA, Lausanne, Switzerland). The uncropped western blot scans were shown in Supplementary Fig. 7.

**Generation of the lentiviral system.** The lentiviral-based vector pLL3.7 was used to express shRNAs in hepatocellular carcinoma cells. The oligonucleotides (Invitrogen) were annealed and subcloned into pLL3.7. Lentiviruses were generated by transfecting subconfluent HEK293T cells together with the lentiviral vector and packaging plasmids by calcium phosphate transfection. Viral supernatants were collected 48 h after the transfection, centrifuged at 75,000g for 90 min, resuspended and filtered through 0.45 μm filters (Millipore). Freshly plated hepatocellular carcinoma cells were infected with the lentivirus, and the knockdown efficiency for the target genes was determined by western blot or RT-PCR. The oligonucleotide sequences for shRNA targeted mRNA(5′ to 3′) were GGGCATGGTG AAGGAAGTT for Nur77, GACCATCCAGAAGAACACAAT for PEPCK1, CTTCACAATTCCGAGACAT for p300 and GGTGTGACTAACTATGCAA for Snail.

**Colony formation and cell proliferation assay.** For colony formation assay, cells were seeded in 60 mm dishes and cultured at 37 °C and 5% CO$_2$ for 15 days. Colonies were fixed with methanol and stained with 0.1% crystal violet in 20% methanol for 15 min and then photographed. For cell proliferation, cells were seeded in triplicate in each well of a six-well plate, and the cell numbers were counted every day over a 6-day period.

**Quantitative real-time PCR.** Total RNA was extracted using Trizol (Invitrogen), and complementary DNA was synthesized using FastQuant RT Kit (TIANGEN, Beijing, China). Specific quantitative real-time PCR experiments were performed using SYBR Green Power Master Mix following manufacturer's protocol (Promega, Madison, USA). Specific primers used for qRT-PCR assays were 5′-CTCTGGAGGTCATCCGCAAG-3′, 5′-CTGGCTTAGACCTGTACGC C-3′ for Nur77; 5′-GCTCTGAGGAGGAGAATGG-3′, 5′-TGCTCTTGGGTGA CGATAAC-3′ for PEPCK1; 5′-GAGGTCACTTCTGAGGAGGAG-3′, 5′-GCTG AGTAGAGACTGGCTGG-3′ for E2F1; 5′-AATCCTTTCCATACCGAACC-3′, 5′-GAGGGCAGTCAGAGCCATAC-3′ for p300; 5′-ATTATCCATCTTCGCC ACCA-3′, 5′-CTCTGCTTGAGCTGGGTCTT-3′ for Ubc9; 5′-GGTGGACA AGGATGTGAAGATA-3′, 5′-GGGAACTTCTTCCTCTGGATG-3′ for FBP1; 5′-ACCCGGCTCTTGGTGAATTT-3′, 5′-ATATTCAGTGGTGGCCGCAT-3′ for FBP2; 5′-CGCAATGGGAAGTACGATCT-3′, 5′-TCGATGGAGACCACAG GATA-3′ for ENO3; 5′-GGGAAAGATAAAGCCGACCTAC-3′, 5′-CAGCAAG GTAGATTCGTGACAG-3′ for G6Pc; 5′-CAGCCTTCCTTCCTGGGCATG-3′, 5′-ATTGTGCTGGGTGCCAGGGCAG-3′ for β-actin.

**Luciferase reporter assay.** Cells were transfected with various plasmids, including a luciferase-linked reporter gene, a β-galactosidase (β-gal) expression vector and other vectors as required. After transfection, cells were lysed and measured for luciferase and β-gal activities. The β-gal activity was used to normalize for transfection efficiency.

**Preparation of subcellular fractionation.** Subcellular fractionation was performed as previously described[20]. Briefly, after washing with PBS, cell were scraped and lysed in 0.5 ml of hypotonic NP-40 buffer (10 mM Hepes (pH 7.9), 10 mM KCl, 0.15% NP-40, 0.1 mM EDTA (pH 8.0) and 0.1 mM EGTA and protease inhibitors) on ice for 10 min. The cell lysates were centrifuged at

4,000 rpm for 5 min. The supernatant (cytoplasmic fraction) was collected. The nuclei-containing pellet was washed twice with cold hypotonic NP-40 buffer and resuspended in SDS buffer (50 mM Tris-HCl (pH 8.0), 10 mM EDTA and 1% SDS and protease inhibitors) for sonication to obtain the nuclear fraction.

**ChIP assay.** Cells were cross-linked by adding formaldehyde directly to culture medium to a final concentration of 1% and incubated for 10 min at 37 °C. Pellets were suspended in 200 μl of SDS lysis buffer (1% SDS, 10 mM EDTA and 50 mM Tris (pH 8.1)), then sonicated. After centrifugation, soluble chromatin was subjected to immunoprecipitation using certain antibodies. DNA was recovered by phenol/chloroform extraction and analysed by PCR using primers against relevant promoters.

**Sumoylation and ubiquitination assays.** To detect PEPCK1-SUMO1 *in vivo*, precleared cell lysates containing N-ethylmaleimide (NEM, 20 mM), a potent inhibitor of de-sumoylation enzymes, were rotated with anti-PEPCK1 or anti-SUMO1 antibody at 4 °C for 4 h. Immunoprecipitates, collected on protein G-Sepharose, were washed three times with ELB lysis buffer and then subjected to western blot. For sumoylated-PEPCK1 detection in cells that were transfected with Flag-PEPCK1, HA-Ubc9 and Myc-SUMO1, cell lysates containing NEM were incubated with anti-Flag antibody, precipitated with protein G-Sepharose, washed three times with ELB and analysed by western blot.

For ubiquitination assay, cell lysates were subjected to immunoprecipitation with anti-Flag antibody, eluted by boiling 5 min in 1% SDS, diluted 10 times in ELB lysis buffer and then re-immunoprecipitated with anti-Flag antibody. The ubiquitin-conjugated proteins were detected by western blot.

**DNA methylation analysis.** Genomic DNA was treated with sodium bisulfite using the EpiTect system (Qiagen Gmbh, Hilden, Germany). Briefly, 2 μg DNA in 20 μl volume was used for each reaction and mixed with 85 μl bisulfite and 35 μl DNA protect buffer. The bisulfite-treated DNA was recovered by EpiTect spin column eluted with 20 μl Buffer EB. Then, completely methylated and unmethylated control DNA samples were tested for each primer pair. PCR products were analysed after electrophoresis on 2% agarose gels containing ethidium bromide.

**Metabolic assays.** For glucose uptake assay, cells were seeded in six-well plate, and then treated with 30 nM 2-(N-(7-Nitrobenz-2-oxa-1,3-diazol-4-yl)Amino)-2-Deoxyglucose (2-NBDG) for 1 h at 37 °C. The uptake of 2-NBDG was analysed by flow cytometry.

To measure glucose output, cells were seeded in six-well plate. The medium was replaced with 1 ml of glucose-free DMEM supplemented with 2 mM sodium pyruvate and 20 mM sodium lactate. After 3 h incubation, medium was collected and the glucose concentration was measured with Glucose Colorimetric/Fluorometric Assay Kit (Bio vision). The absorbance of the supernatant was measured at Ex/Em = 535/587 nm with POLARstar Omega (BMG Labtech, VIC, Australia).

OCR and ECAR were measured using a Seahorse Biosciences XF96 analyzer (North Billerica, MA). Cells were cultured in plate for 24 h, and then acclimatized at 37 °C for 2 h in XF media (non-buffered DMEM containing 25 mM glucose and 2 mM glutamine). OCR was measured using XF Cell Mito Stress Test Profile according to the manufacture's handbook. Briefly, basal OCR were measured under the basal condition followed by the sequential addition each well with oligomycin (3.5 μM), carbonyl cyanide-ptrifluoromethoxyphenylhydrazone (2 μM), and Rotenone/antimycin A (1 μM). ECAR was measured using XF Glycolysis Stress Test Profile according to the manufacture's handbook, and basal ECAR were measured under the basal condition followed by the sequential addition each well with glucose (10 mM), oligomycin (3.5 μM) and 2-deoxyglucose (100 mM) to the indicated final concentrations. XF assay is consisted of sequential mix (3 min), pause (3 min), and measurement (5 min) cycles, allowing for determination of OCR/ECAR every 10 min. The ECAR bar values represent for the glycolytic capacity, which is got by subtracting the non-glycolytic acidification from the maximum values. The OCR bar values represent for the ATP production-related oxygen consumption, which is subtracting out non-mitochondrial respiration and proton leak from the basal values.

**Detection of ATP level.** Intracellular ATP level was assayed by a bioluminescence method with the ENLITEN ATP Assay System Bioluminescence Detection Kit (Promega) according to the kit instructions. Briefly, cells (2 × 10$^4$) were seeded in six-well plates for 24 h and then collected and homogenized with 500 μl of ice-cold homogenization buffer (0.25 M sucrose and 10 mM HEPES-NaOH (pH 7.4)). Equal volume of ice-cold 10% TCA (trichloroacetic acid) was added to homogenate and shaken for 20 s and then centrifuged at 10,000g for 10 min at 4 °C. Supernatant (400 μl) was collected and added to 200 μl of Tris-acetate buffer (1 M, pH 7.75) for neutralization. The supernatant was diluted 50-fold with deionized water, and 20 μl of the diluted extract was used for ATP assay by a luminometer.

**Identification of interaction proteins and sumoylated sites.** For detecting the interaction protein with Nur77, Flag-Nur77 was transfected into 293T cells, and then immunoprecipitated with anti-Flag antibody and eluted with $200\,\mu g\,ml^{-1}$ Flag peptide (Sigma). The proteins were separated by SDS-PAGE followed by silver staining. Visible band from silver-stained gel was cut out, and proteins were in-gel digested with trypsin, and analysed by liquid chromatography-mass spectrometry.

To detect the PEPCK1 sumoylation sites, 293T cells that transfected with Flag-PEPCK, HA-UBC9 and Myc-SUMO-1(RGG) were incubated with anti-Flag antibody. Purified PEPCK1 proteins were separated by 8% SDS-PAGE, and then stained with coomassie blue. The SUMO1-PEPCK1 conjugates were excised in separate gel slices. Proteins were digested in-gel with trypsin. The tryptic peptides were analysed on an AB Sciex TripleTOF 5600 mass spectrometer interfaced to an Eksigent NanoLC Ultra 2D Plus HPLC system. The interpretation of both the MS and MS/MS data were carried out with ProteinPilot V4.5 beta. Spectra were inspected manually to eliminate false positives, excluding spectra with low S/N, erroneous modification assignments and confidence values below 95% unless justifiable by the presence of a-ions after comparison with the theoretical MS/MS product ion spectrum.

**Statistical analysis.** All the statistical analyses were performed by GraphPad Prism 5. The statistical data are represented as the mean ± s.e.m. The statistical analysis of different groups is realized using the two-tailed Student's $t$-test or one-way ANOVA followed by Tukey post hoc test. The correlations of relative immunoreactive score (IRS) of Nur77 and PEPCK1, Nur77 and PCNA, Nur77 and nuclear Snail in clinical HCC tissue were analysed using Pearson's chi-squared test. Kaplan-Meier survival analysis was used to evaluate the relationship between Nur77 or PEPCK1 expression and HCC prognosis. The significance thresholds were deemed as statistically significant ($P \leq 0.05$), highly significant ($P \leq 0.01$) or extremely significant ($P \leq 0.001$).

**Data availability.** All relevant data that support the findings of this study are available within the article and its Supplementary Information Files or from the corresponding author on request.

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

## Acknowledgements

We thank that Dr Olle Larsson (Karolinska Institute, Stockholm, Sweden) kindly provides HA-UBC9 and Myc-SUMO-1 plasmids, and Flag-Snail is a gift from Dr Chundong Yu (Xiamen University, Fujian, China). Dr Shi-min Zhao (Fudan University, Shanghai, China) provides anti-acetylated lysine antibody for immunoprecipitation of acetylated proteins. This work was supported by grants from the National Natural Science Fund of China (U1405224), the Ministry of Science and Technology and the National Natural Science Fund of China (2014CB910602), the Programme of Introducing Talents of Discipline to Universities (111 Project, B12001), Open Research Fund of State Key Laboratory of Cellular Stress Biology, Xiamen University (SKLCSB2016KF001).

## Author contributions

Q.W. and H.Z.C. conceived the study, generated hypotheses and designed experiments. X.L.B., H.Z.C., P.B.Y., Y.P.L., F.N.Z., J.Y.Z. and W.J.W. performed the cell, animal and clinical experiments with data analysis; W.X.Z., S.Z., and X.M.W. provided clinical samples; Q.T.C., Y.Z. and X.Y.S. constructed various plasmids; K.Y.J. performed experiment of mass spectrometry with data analysis; Q.W. wrote the paper.

## Additional information

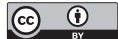

