## [Peer Review File · Nature Communications]

Reviewers' comments:

Reviewer #1 (Remarks to the Author):

The authors present a compelling manuscript that Nur77 plays an important role in HCC via regulation of PEPCK sumoylation. A quite comprehensive set of data is presented from clinical samples to detailed biochemical mechanistic studies. The sumoylation of PEPCK is a novel finding. The authors go further, characterizing the machinery mediating this effect as well as the role of Nur77 and p300 in regulating this process, as well as how Nur77 is downregulated in HCC. Finally, they propose that the ability of PEPCK to promote HCC and cell line growth via regulation of glucose metabolism. Despite a tremendous amount of data, there are serious technical, experimental and presentations concerns that strongly diminish enthusiasm. Conflicting data with the literature, and experiments that are confusing further reduce enthusiasm.

The data showing increased DEN induced HCC in Nur77 knockout mice is compelling. However, the authors also show that HFD also induces HCC in Nur77 KO mice vs wt mice (supplemental Figure 1B). There are a number of issues with this experiment.

Previous studies using Nur77 knockout mice show that HFD induces increased steatosis (Chao et al 2009). Furthermore, HFD increases HCC. However, the authors do not observe this effect. There are more tumors in control fed mice, whereas HFD usually induces HCC more than control diets. While these are different experiments, and hence different results, HFD increases the number, latency and size of liver tumors in general. In addition, the livers of KO mice on HFD do not look steatotic which is usually observed in these mice especially under H&E as reported by Chao et al 2009. Furthermore, insulin levels are elevated in Nur77 KO HFD mice. Insulin (and IGF) could be mediating many of the observed effects. This needs to be ruled in and out under control and HFD conditions.

The authors need to show Nur77 and PEPCK expression in the HCC tumors at RNA and protein level. It would also be useful to show Ubc9 and p300 levels.

Using limited cell lines the authors claim that Nur77 is lower in HCC cell lines. However, HepG2 have similar levels and LO2.

Previous work shows that PEPCK promotes the TCA cycle, this would lead to increased NADH and FADH₂, and hence ETC and oxygen consumption (Burgess et al 2004). Indeed, overexpression of PEPCK in muscle increases the oxidative capacity of the muscle (Hanson and Hakimi 2008). The authors need to reconcile these discrepancies .

The authors claim in methods that they examined glycolytic capacity. However, the technique they used appears to be the mitochondrial stress test, which is not the protocol for glycolytic capacity. Rather, the glycolytic stress is usually used. At what point are the authors measuring basal ECAR and OCR. Is this subtracting out non-mitochondrial respiration, or simply the initial ECAR and OCR prior to oligomycin injection. Perhaps the authors should show the Seahorse tracing.

PEPCK is elevated in the liver of obese mice. This would seem to be at odds with the data presented. Therefore, PEPCK should lead to protection against HCC?

Almost all the data showing sumoylated PEPCK is following immunoprecipitation of endogenous PEPCK and blotting for PEPCK or tagged PEPCK and blotting for tag. What about endogenous PEPCK under conditions of increased sumoylation. Most of the endogenous PEPCK blotting shows changes in PEPCK by Nur77 or curcumin. However, differences are difficult to discern perhaps because figures are difficult to read (see below).

Although the effect of Nur77 on clonogenic survival is not dependent on its transcriptional activity, it does not rule out an effect of Nur77 on PEPCK gene expression. Furthermore the large induction

of Nur77 leads to only a small increase in PEPCK. It might be that the large overexpression leads to increased nuclear Nur77 and hence transcriptional activation of PEPCK. This needs to be addressed. Perhaps also use the zinc finger binding mutant in the presence of PEPCK shRNA (similar to figure 2G). In addition, what happens to the protein and RNA expression of the other Nur77 targetes

The data on curcumin is concerning. It is unclear why it was included as it really does not add to the story and actually is quite contradictory. Numerous studies have shown that curcumin has antidiabetic, antigluconeogenic activity. This is odds with the data shown.

HepG2 are considered to be non tumorigenic. However, the authors show xenograft data with these cells using 2X10⁶ cells. More details need to be provided as to how they were able to get these cells to form xenografts.

Huh7 in general do not express PEPCK protein (although there is abundant RNA). Therefore it is unclear how the authors were able to show PEPCK in this cell line and especially the changes observed. Indeed, in several figures, PEPCK expression is absent in these cells, eg. Supplemental figure 2F vs 2G. While the overexpression might explain some of this data, this is problematic.

According to the data, there is no PEPCK in HCC tumor tissue. Based on work done several decades ago, while perhaps decreased, there is PEPCK in liver. This is observed in the Oncomine and TCGA as well.

The authors do not discuss the use of NEM. If it is to show the sumoylation of PEPCK, why is it not used in other figures. In addition, how biologically relevant is such a small amount of sumoylation in light of the large amount of unsumoylated PEPCK. Indeed, the decreases shown in figure 3D are difficult to see, although this could be a result of such small figures (See note below regarding figures). In addition, the lower panel of figure D cannot be compared to the upper figure since Ubc and Sumo were not coexpressed. Rather anacardic acid should have been included with Sumo and Ubc9.

The experiments showing PEPCK inhibits cell proliferation should also be performed as a xenograft experiment.

Almost all the experiments showing sumoylation of PEPCK are following IP, with the exception of supplementary figure 5A and 5B, in which PEPCK was overexpressed (in a cell with PEPCK). Therefore, can the sumoylation be observed in cells without ectopic expression of PEPCK. Also can sumoylated PEPCK be observed in the absence of pulling down one of the transfected proteins. What about blotting for endogenous PEPCK vs the flag epitope with and without IP. Indeed in figure 3C, the authors A) show no change in Flag PEPCK in lysate, but also cut the blot so it cannot be determined whether there is sumoylated PEPCK in these lysates.

A more clear idea of control of PEPCK expression by Nur77 over time without CHX would be more informative (similar to figure 5A, but without CHX, not just time zero, indeed, although the blots cannot be compared since they don't look they are on the same blot, it does not look like there is a difference or if anything, PEPCK levels seem lower in cells with Nur77 at time 0.

Other issues:

Reviewing the manuscript was very difficult since the size of the figures was so small. In addition, the manuscript is need of significant copy editing.

A number of statements that are made are overly stated

Eg. "anacardic acid markedly stimulated..." there was a small change- not markedly.

It is unclear how the authors can make the statement on line 215. Both ALLN and MG132 will inhibit proteosomal degradation. Therefore, this raises the questions as to how ALLN did not have a similar effect as MG132.

Reviewer #2 (Remarks to the Author):

In the here present study authors evaluate importance of the nuclear receptor Nur77 and PEPCK1 enzyme as potential tumor suppressors in hepatocellular carcinoma (HCC). By utilizing different approaches and models (in vitro, in vivo, human samples) the authors aim to explore the effects of Nur77 downregulation on HCC development. Further, they delineate a new regulatory role for Nur77 for PEPCK1 sumoylation and subsequent ubiquitination.

Key findings:

- Majority of the HCC patients show downregulation of Nur77 in the tumor tissue compared to the normal liver. Patient expressing low amount of Nur77 have close association with poor clinical prognosis.
- Increased level of Nur77 induces tumor suppressive effect in HCC by promoting gluconeogenesis, preventing glucose uptake and depleting intracellular ATP.
- Nur77 prevents sumoylation and degradation of PEPCK1, a rate-limiting enzyme of gluconeogenesis. Stability of PEPCK1 is dependent on sumoylation of Lys124.
- Curcumin induces Nur77 gene and protein upregulation and, therefore, promotes antitumorigenic properties.
- Snail transcription factor is suppressing Nur77 gene expression by binding to E-boxes on Nur77 promoter.

Overall, the study focuses on an important topic, namely the relevance of key regulators of gluconeogenesis for hepatocarcinogenesis. The study is technically well performed. However, some of the mechanistic findings need to be extended and utilized cell lines should be consistently applied across the experiments. A potential p53-dependent role should further be explored (Zhang et al. PNAS 2014).

Major comments

- Given the impact of PEPCK1 on T-cell function (Ho et al. Cell 2015) and the importance of the chronic inflammatory liver disease for HCC development, pre-neoplastic stages of hepatocarcinogenesis should be explored in the context of the here suggested study. Further, clinic-pathological information for the investigated specimens should be added. Further, the data for the Nur77 expression in the investigated HCC cohort should be shown.
- Throughout the manuscript the utilized of cell lines are inconsistently applied (e.g. Figure 1 and 2). This should be extended, in particular for the investigation where only hepatoma cell line (and 293T cells) was used (e.g. Figure 4). Further, it would be interesting to include untransformed hepatocytes as a negative control. Furthermore, given the dominant role of p53 (also suggested in the introduction) on gluconeogenesis and the high incidence of p53 mutations in HCC, it might be interesting to explore this issue. This could be achieved, e.g. by adding the p53-null cell line

Hep3B. Of note, SK-Hep1 is a cell line of endothelial origin (Heffelfinger et al. 1992) and should not be used synonymously in the context with HCC cell line. The cell line should be replaced.

- The authors suggest a direct regulatory effect of Nur77 on PEPCK1 (Supplemental Figure 2). While this is certainly interesting, mechanistic investigations confirming the suggested direct regulatory effect are missing. This should be extended to confirm the Nur77-PEPCK1 regulatory role for HCC. Further, despite the potential regulation of PEPCK1 it remains unclear how Nur77 affects the tumorigenic potential of HCC cells, e.g. by regulating WNT signaling (Chen et al. Gut 2012). In line with this, impact of Nur77 on sumoylation and the suggested mechanism of operation are not striking and most pronounced in non-hepatoma 293T cells (Figure 5). Quantification and statistical evaluation is needed. Similar, statistical analyses of (e.g. Supplemental Figure 5A) seem necessary, since different base line levels of PEPCK1 expression are shown.
- Given the plethora of different curcumin targets, the mechanisms of Nur77 upregulation by curcumin are unclear and should be evaluated. Since curcumin is known to inhibit cell growth of hepatoma cells by affecting multiple molecular targets, specificity for the suggested findings should be confirmed by a more targeted approach.
- Promotor methylation of Nur77 should be confirmed in the cell lines (Figure 6). Rational for Snail selection should be explained in more detail.

Minor comments

- The study by Wurmbach et al. includes 4 stages of HCC. Please explain the differences in the here presented study. Please also explain how the statistical analyses shown in the Figure have been derived.
- The effect of different Nur77 levels on proliferative status in individual tumors (e.g. by Ki67 and PCNA) should be confirmed.
- The authors suggest that Nur77 is driven by non-genomic events. Please extend the explanation of this hypothesis. Maybe data supporting that Nur77 is not impaired by genetic alterations (e.g. by using publically available repositories) can be included.
- It is unclear how the doses for the used compounds were established in the respective cell lines. Experimental data or suitable references should be provided.
- Number of replicates and statistical analyses should be added wherever necessary.

Reviewer #3 (Remarks to the Author):

Bian et al's manuscript demonstrates how Nur77 as a tumor suppressor suppresses hepatocellular carcinoma. They show that Nur77 is a tumor suppressor of HCC and affects gluconeogenesis via interacting with PEPCK1. They also show that PEPCK1 is a SUMOylated protein and SUMOylation affects PEPCK1 stability. More interestingly, they find that p300 enhances PEPCK1 sumoylation through acetylating UBC9. Increasing Nur77 expression by compound curcumin could suppress PEPCK1 sumoylation and HCC. This manuscript is quite complicated but of interesting, especially they find that p300 acetylation of UBC9 enhances PEPCK1 sumoylation, which will be a novel regulation mechanism for sumoylation.

Major:

1. Authors showed that sumoylation down-regulated PEPCK1 stability, which would decrease gluconeogenesis and increase tumorigenesis in HCC. If it is true, it should be detected that PEPCK1 is highly-sumoylated in HCC samples compared to in normal liver tissue, which would strongly support the link between PEPCK1 SUMOylation and tumor suppressor function of Nur 77. Additionally, how sumoylation affect PEPCK1 stability?
2. It has been shown that sumoylation of PEPCK1 decreases tumorigenesis in HCC. Therefore, sumoylation mutant of PEPCK1 would rescue the tumorigenesis in Nur77-silenced HCC. They should do that to conclusively make statement that Nur 77 suppress HCC via PEPCK1 sumoylation.

3. It is of interesting that p300 acetylation UBC9 increases PEPCK1 sumoylation. Does p300-UBC9 regulation affect global sumoylation? It seems that acetylated UBC9 has more affinity to bind to PEPCK1 than un-acetylated form. They should prove it by binding experiment.

4. The authors mention that hepatic expression of Nur77 can elevate gluconeogenesis through transcriptional upregulation of gluconeogenic genes, including G6pc, Fbp1 and Fbp2, and Eno317. They also show that mutation of Nur77 binding to DNA affect the growth of SMMC-7721 cells (Supplemental Figure 2B), suggesting that the transcription activity of Nur77 also engage the tumorigenesis. How to comment the contribution of Nur77-PEPCK1 non-transcription activity and Nur77 transcription activity to HCC?

Minor:

1. In Fig 3a, Is there any changes in PEPCK1 sumoylation in liver with or without DEN/CCL4 treatment?

2. In Fig 4, Does knock-down p300 affect PEPCK1 expression as well as sumoylation in HCC cells?

3. In Fig 4a (bottom), overexpression of p300 decreased endogenous PEPCK1 expression in 293T and HepG2, did affect the sumoylation of endogenous PEPCK1?

4. The author mentioned in the discussion that the third mechanism of Nur77 regulating PEPCK1 is attenuating p300-induced PEPCK1 acetylation, is there any evidence for it?

Responses to Reviewer #1

1. Previous studies using Nur77 knockout mice show that HFD induces increased steatosis (Chao et al 2009). Furthermore, HFD increases HCC. However, the authors do not observe this effect. There are more tumors in control fed mice, whereas HFD usually induces HCC more than control diets. While these are different experiments, and hence different results, HFD increases the number, latency and size of liver tumors in general. In addition, the livers of KO mice on HFD do not look steatotic which is usually observed in these mice especially under H&E as reported by Chao et al 2009. Furthermore, insulin levels are elevated in Nur77 KO HFD mice. Insulin (and IGF) could be mediating many of the observed effects. This needs to be ruled in and out under control and HFD conditions.

HFD can promote HCC development (*Cell*. 2010;140(2):197-208). This effect of a HFD was observed by comparing HFD and normal chow in the same HCC mouse model. However, in our study, we used two different HCC mouse models, DEN/CCl₄- and HFD/STZ-induced hepatocarcinoma models, which mimic the different pathogenesis of human HCC, to evaluate the effect of Nur77 on HCC tumor growth (Figure 1d & Supplementary Figure 1f). Therefore, it is not suitable to compare tumor size and number between these two different models. To avoid misunderstanding, we have added a schematic overview for the two methods of HCC induction in Figure 1d & Supplementary Figure 1f.

Consistent with the results reported by Chao *et al* (*Diabetes*. 2009; 58(12):2788-96), HFD indeed induced steatosis, which was substantially more severe in Nur77-KO mice, as indicated by H&E staining in Supplementary Fig. 1e.

Although insulin levels are elevated in Nur77-KO-HFD mice (*Diabetes*. 2009; 58(12): 2788-96), the following results indicated that insulin or IGF signaling was not involved in Nur77-mediated HCC suppression. In the DEN/CCl₄ model, Nur77-KO mice developed HCC at a higher rate than WT mice; however, insulin and IGF1 levels were equal between these groups of mice (Supplementary Fig. 1h). Similarly, in the HFD/STZ model, in which mice were intraperitoneally injected with the β cell toxin streptozotocin (STZ) at 2 days of age, the insulin and IGF1 levels in Nur77-KO mice were also comparable with those in WT mice (Supplementary Fig. 1h), unlike the reported HFD model (i.e., no injection with STZ. *Diabetes*. 2009; 58(12): 2788-96). Together, our results suggest that Nur77 suppresses HCC independent of insulin or IGF1.

2. The authors need to show Nur77 and PEPCK expression in the HCC tumors at RNA and

protein level. It would also be useful to show Ubc9 and p300 levels.

Nur77 and PEPCK mRNA levels and Ubc9 and p300 mRNA and protein levels in clinical HCC samples are now provided in Figure 1a, Figure 2b and Supplemental Figure 4i-j, respectively.

3. Using limited cell lines the authors claim that Nur77 is lower in HCC cell lines. However, HepG2 have similar levels and LO2.

To further verify the lower expression of Nur77 in HCC cell lines, we used three additional liver cancer cell lines and displayed their mRNA and protein expression levels in Supplementary Fig. 1i. Consistent with our previous results, Nur77 expression was substantially lower in HCC cells than in the untransformed hepatocyte L02 cells. Although HepG2 and BEL-7402 expressed relatively higher Nur77 levels compared with other HCC cells, Nur77 mRNA and protein levels in these two cells were still lower than in L02 cells.

According to the suggestion from Reviewer #2, we removed the Sk-hep-1 cell line. The Sk-hep-1 cell line is derived from the ascitic fluid of a patient with adenocarcinoma of the liver; it is a cell line of endothelial origin (*In Vitro Cell Dev Biol.* 199;28A(2):136-42). Therefore, it is not suitable to be used in conjunction with HCC cell lines.

4. Previous work shows that PEPCK promotes the TCA cycle, this would lead to increased NADH and FADH2, and hence ETC and oxygen consumption (Burgess et al 2004). Indeed, overexpression of PEPCK in muscle increases the oxidative capacity of the muscle (Hanson and Hakimi 2008). The authors need to reconcile these discrepancies .

Burgess and colleagues reported that liver-specific knockout of PEPCK1 in mice induces the accumulation of TCA cycle intermediates, which results in inhibition of the TCA cycle and oxygen consumption (*J Biol Chem.* 2004; 279(47):48941-9). Hakimi and colleagues reported that overexpression of PEPCK1 in skeletal muscle removes TCA cycle intermediates to ensure that the high rates of flux required for supporting exercise continues (*Biochimie.* 2008; 90(6):838-42; *J Biol Chem.* 2007; 282(45):32844-55). These results are consistent with the notion that PEPCK1 is a major cataplerotic enzyme that removes TCA cycle intermediates by converting oxaloacetate to PEP (*J Biol Chem.* 2002; 277(34):30409-12). However, this function of PEPCK1 on TCA cycle flux may not hold true for cancer cells. Due to the high proliferative activity of cancer cells, TCA cycle intermediates are usually shifted for macromolecular synthesis, which is critical to supply enough nucleotides, proteins, and lipids for cell doubling (*Cancer Cell.* 2012; 21(3):297-308). Therefore, PEPCK1 knockdown in

cancer cells may not accumulate TCA cycle intermediates, which is also reported by another paper (*Mol Cell*. 2015; 60(4):571-83). In our study, PEPCK1 knockdown in HCC cells leads to the inhibition of glucose production and the promotion of glucose utilization, which finally results in enhanced oxygen consumption in the TCA cycle (Supplementary Fig. 21). This interpretation has been added to the Discussion section.

5. The authors claim in methods that they examined glycolytic capacity. However, the technique they used appears to be the mitochondrial stress test, which is not the protocol for glycolytic capacity. Rather, the glycolytic stress is usually used. At what point are the authors measuring basal ECAR and OCR. Is this subtracting out non-mitochondrial respiration, or simply the initial ECAR and OCR prior to oligomycin injection. Perhaps the authors should show the Seahorse tracing.

We apologize for not clearly explaining the OCR and ECAR measurement procedure. In fact, we have performed the glycolysis stress test for ECAR measurement and the mitochondrial stress test for OCR measurement. The ECAR bar values represent the glycolytic capacity, which is attained by subtracting the non-glycolytic acidification from the maximum values. The OCR bar values represent the ATP production-related oxygen consumption, subtracting out non-mitochondrial respiration and proton leak from the basal values. The seahorse tracing curves are now shown in Supplemental Figures 2a, 2m, and 2o. The procedure for OCR and ECAR measurement has also been rewritten in the Methods section.

6. PEPCK is elevated in the liver of obese mice. This would seem to be at odds with the data presented. Therefore, PEPCK should lead to protection against HCC?

It has been reported that PEPCK1 is elevated in the liver of obese mice (*J Biol Chem*. 2014; 289(6):3244-61; *J Mol Endocrinol*. 2013; 50(2):167-78). However, this finding cannot preclude the anti-tumorigenic effect of PEPCK1 against HCC. The effect of PEPCK1 against HCC should be verified directly by comparing tumor development between WT and PEPCK1-KO mice. Unfortunately, we do not have these mice.

Tumor initiation and progression involve multiple steps that are regulated by multiple signaling pathways. One single gene may not consistently influence each step. For example, p53, a well-known tumor suppressor gene, is also reported to be activated in the liver of obese mice or alcoholic fatty liver of rat (*J Biol Chem*. 2004; 279(20):20571-5; *J Hepatol*. 2011; 54(1):164-72), although obesity and alcoholic liver disease are considered the main risk

factors for HCC. Similarly, although the elevation of PEPCK1 in obese mice may contribute to high blood glucose, PEPCK1 also functions as a barrier for HCC development, as suggested by our study. Our results clearly indicated that PEPCK1 knockdown promoted HCC proliferation (Figure 2i). In contrast, PEPCK1 overexpression inhibited tumor growth at both the cellular level and mouse level (Figure 2f, Supplemental Figure 2p). Clinical patients with lower PEPCK1 expression were associated with poor clinical prognosis (Figure 2d). Therefore, enhanced PEPCK should protect against HCC.

7. Almost all the data showing sumoylated PEPCK is following immunoprecipitation of endogenous PEPCK and blotting for PEPCK or tagged PEPCK and blotting for tag. What about endogenous PEPCK under conditions of increased sumoylation. Most of the endogenous PEPCK blotting shows changes in PEPCK by Nur77 or curcumin. However, differences are difficult to discern perhaps because figures are difficult to read (see below).

To answer the question of “What about endogenous PEPCK under conditions of increased sumoylation,” we added an experiment using HepG2 cells, in which the endogenous sumoylation of PEPCK1 could also be detected in the presence of NEM, an inhibitor of de-sumoylation (Figure 3c). NEM was thus used to detect PEPCK1 sumoylation hereafter. We further verified that in different HCC cell lines, increased endogenous sumoylation of PEPCK1 by transfection of p300 was associated with decreased endogenous PEPCK1 protein level (Figure 4b). In clinical HCC samples, increased sumoylation accompanied decreased PEPCK1. Conversely, there is no sumoylation, but high PEPCK1 levels, in para-carcinoma samples (Figure 3b). Thus, higher sumoylation indeed corresponds to lower endogenous PEPCK1 levels *in vivo*.

In our study, PEPCK1 sumoylation promoted its ubiquitin-proteasomal degradation (Figure 3h), suggesting that endogenous sumoylated PEPCK1 is more labile and difficult to detect without immunoprecipitation.

Unlike other modifications, such as ubiquitination, sumoylation is described as “low-level, big-effect.” Few proteins are quantitatively sumoylated, either constitutively or upon receiving their respective upstream signals (*J. Proteome Res.* 2015, 14:2385). Instead, most targets appear to be modified to a small percentage at steady state. However, the fact that a small pool of sumoylated protein causes the dramatic effects that have been assigned to sumoylation has been well recognized (*EMBO J.* 2002, 21:1456–64; *Mol. Cell.* 2003, 11:1043–1054; *Mol. Cell.* 2006, 24:341–354). As Ruth Geiss-Friedlander and Frauke Melchior mentioned in their review (*Nature Reviews Molecular Cell Biology.* 2007,

8:947-956), sumoylated targets can undergo rapid cycles of modification and demodification. Although this equilibrium might lie on the side of the unmodified form, the whole pool of a given protein might be affected by sumoylation in a short window of time (*Cell Metabolism*. 2014, 20: 603–613). For example, steady-state sumoylation is usually less than 5%, and most transcription factors and co-regulators become significantly activated when the SUMO acceptor Lys residue has been mutated to an Arg residue (*Mol. Cell*. 2014, 13: 611–617). Therefore, in most reports, the sumoylation level (particularly the endogenous level) is lower, but its outcomes are extremely diverse and important, including the localization of proteins and the activities of enzymes and target proteins.

8. Although the effect of Nur77 on clonogenic survival is not dependent on its transcriptional activity, it does not rule out an effect of Nur77 on PEPCK gene expression. Furthermore the large induction of Nur77 leads to only a small increase in PEPCK. It might be that the large overexpression leads to increased nuclear Nur77 and hence transcriptional activation of PEPCK. This needs to be addressed. Perhaps also use the zinc finger binding mutant in the presence of PEPCK shRNA (similar to figure 2G). In addition, what happens to the protein and RNA expression of the other Nur77 targetes.

Thank you for your questions and suggestion.

PEPCK1 promoter sequence analysis revealed that unlike other gluconeogenic genes, there is no typical Nur77 response element (NurRE) or NGFI-B response element (NBRE) on the PEPCK1 promoter. It is unlikely that Nur77 directly binds to PEPCK1 promoter and activates its transcription. Transfection of Nur77 at increasing doses did not affect PEPCK1 gene expression in SMMC-7721, HepG2, or Huh7 HCC cells or untransformed hepatocyte L02 cells (Supplemental Figure 5b). These results rule out the possibility that “the large overexpression leads to increased nuclear Nur77 and hence transcriptional activation of PEPCK” and confirms that Nur77 regulation of PEPCK1 occurs in a post-transcriptional manner.

Furthermore, Nur77 but not its zinc finger binding mutant (Nur77 2G) elevated the mRNA and protein level of E2F1, a downstream target of Nur77 (*J Biol Chem*. 2003; 278(44):42840-5), in untransformed hepatocyte L02 cells but not in other several HCC cells (Supplemental Figure 2e). The different functions of Nur77 on its downstream target between HCC cells and untransformed hepatocytes may be attributed to the different subcellular localizations of Nur77. In L02 cells, Nur77 is mainly located in nucleus, while Nur77 is mainly distributed in the cytoplasm in HCC cell lines (Supplemental Figure 2d). In addition,

both Nur77 (Figure 2i) and Nur77 2G (Supplemental Figure 2p) efficiently inhibited clone formation in the control group, but not in PEPCK1-KD Huh7 cells. Together, these results further demonstrate that Nur77 has an effect on cell proliferation through PEPCK1 mediation that is independent of its transcriptional activity.

9. The data on curcumin is concerning. It is unclear why it was included as it really does not add to the story and actually is quite contradictory. Numerous studies have shown that curcumin has antidiabetic, antigluconeogenic activity. This is odds with the data shown.

Curcumin has various activities, including anticancer, antidiabetic, and antigluconeogenic activity. However, its mechanism of action is very complex. Based on the questions from you and Reviewer #2, the Editor has suggested deleting all of the curcumin data from the manuscript. We agree with this suggestion.

10. HepG2 are considered to be non tumorigenic. However, the authors show xenograft data with these cells using 2X106 cells. More details need to be provided as to how they were able to get these cells to form xenografts.

In the ATCC catalog, the HepG2 cell line is reported to be non-tumorigenic; however, many papers showed that HepG2 cells can form xenografts in nude mice (*Journal of Hepatology*. 2009, 50:1132–1141). We could obtain the xenograft by injecting with Matrigel-embedded HepG2 cells. As mentioned previously, in this revised version, we have deleted the curcumin-related data, including the HepG2-formed xenograft. To avoid confusion, we used Huh7 cells for the other xenograft experiments shown in Figure 2f & 5f.

11. Huh7 in general do not express PEPCK protein (although there is abundant RNA). Therefore it is unclear how the authors were able to show PEPCK in this cell line and especially the changes observed. Indeed, in several figures, PEPCK expression is absent in these cells, eg. Supplemental figure 2F vs 2G. While the overexpression might explain some of this data, this is problematic.

Although the PEPCK1 protein level in Huh7 cells is relatively low, we could detect its expression after a longer exposure time by Western blotting using an anti-PEPCK1 antibody (Cell Signaling Technology, Cat. No. 12940S). The PEPCK1 protein expression in Huh7 cells has also been reported in other studies (*Cell Death Discovery*. 2015, 1:15016). The different expression levels of PEPCK1 protein in our various results may result from the different exposure times, rather from PEPCK1 overexpression. In the revised version, we chose the

same image with a longer exposure time to replace the original one (Supplemental Figure 2j & 2k, original 2g).

In Supplemental Figure 2j (original 2f), the PEPCK1 level in control group of Huh7 cells appears to be similar to that of SMMC-7721 cells. This finding is also due to the different exposure times. To avoid misunderstanding, we provided a new image, in which PEPCK1 expression in Huh7 and SMMC-7721 cells was shown on the same film with the same exposure time.

12. According to the data, there is no PEPCK in HCC tumor tissue. Based on work done several decades ago, while perhaps decreased, there is PEPCK in liver. This is observed in the Oncomine and TCGA as well.

PEPCK1 was expressed in clinical HCC tumor tissues but at substantially lower levels than in para-carcinoma tissues. Instead of the original image, we chose to present the same image with a relatively longer exposure time (Figure 2b).

13. authors do no discuss the use of NEM. If it is to show the sumoylation of PEPCK, why is it not used in other figures. In addition, how biologically relevant is such a small amount of sumoylation in light of the large amount of unsumoylated PEPCK. Indeed, the decreases shown in figure 3D are difficult to see, although this could be a result of such small figures (See note below regarding figures). In addition, the lower panel of figure D cannot be compared to the upper figure since Ubc and Sumo were not coexpressed. Rather anacardic acid should have been included with Sumo and Ubc9.

NEM is an inhibitor of de-sumoylation. Because sumoylated targets can undergo rapid cycles of modification and demodification, NEM was used in all sumoylation assays described in the Methods section (Page 6 in Supplementary information). To avoid confusion, we have revised Supplementary Figure 3a (original Figure 3c) and also now describe its role in the Result section (page 11).

With regard to the question “such a small amount of sumoylation in light of the large amount of unsumoylated PEPCK,” please see our previous explanation. Such phenotypes of lower sumoylation have also been observed in other reports (*Cancer Cell*. 2014, 25:748–761; *Cell*. 2007, 131:584–595). In particular, the sumoylated PEPCK1 was more labile and was degraded by the ubiquitination pathway. Thus, sumoylated PEPCK1 is difficult to detect.

According to the suggestion, we performed a new experiment in which Sumo and Ubc9 were first transfected into different cell lines as indicated, and then, cells were treated with

anacardic acid. As expected, anacardic acid treatment obviously increased endogenous PEPCK1 expression levels (Figure 3d, bottom). This result, together with the data in Figure 3d (middle), suggested that inhibition of both endogenous and Sumo/UBC9 overexpression-induced PEPCK1 sumoylation efficiently influenced PEPCK1 protein levels.

14. *The experiments showing PEPCK inhibits cell proliferation should also be performed as a xenograft experiment.*

Thank you for your suggestion. We have performed this xenograft experiment. As shown in Figure 2f, PEPCK1 overexpression significantly repressed xenograft tumor growth, and PCNA and Ki67 expression were also repressed in the xenograft tumor of PEPCK1 transfection (Figure 2g), further suggesting the inhibitory role of PEPCK1 on cell proliferation *in vivo*.

15. *Almost all the experiments showing sumoylation of PEPCK are following IP, with the exception of supplementary figure 5A and 5B, in which PEPCK was overexpressed (in a cell with PEPCK). Therefore, can the sumoylation be observed in cells without ectopic expression of PEPCK. Also can sumoylated PEPCK be observed in the absence of pulling down one of the transfected protein. What about blotting for endogenous PEPCK vs the flag epitope with and without IP. Indeed in figure 3C, the authors A) show no change in Flag PEPCK in lysate, but also cut the blot so it cannot be determined whether there is sumoylated PEPCK in these lysates.*

The sumoylation of endogenous PEPCK1 can be observed in cells (Figure 3c & 4b), mice (Figure 3a), and clinical HCC samples (Figure 3b) following IP without ectopic PEPCK1 expression. As mentioned previously, the sumoylated form of PEPCK1 is substantially more labile, so it is very difficult to detected endogenous PEPCK1 sumoylation in the absence of pulling down PEPCK1 or SUMO1.

The expression level of exogenous Flag-PEPCK1 is substantially stronger than that of endogenous PEPCK1, as revealed using an anti-PEPCK1 antibody (shown below). Overexpression of UBC9 and SUMO1 enhances Flag-PEPCK1 sumoylation, which is easy to detect without IP, as shown in Supplementary Figure 3b & 3c (original Supplementary Figure 3a & 3b).

According to your suggestion, we now include a whole image in Figure 3c (now presented in Supplementary Figure 3a), in which the sumoylated PEPCK1 could also be observed in cell lysates.

16. A more clear idea of control of PEPCK expression by Nur77 over time without CHX would be more informative (similar to figure 5A, but without CHX, not just time zero, indeed, although the blots cannot be compared since they don't look they are on the same blot, it does not look like there is a difference or if anything, PEPCK levels seem lower in cells with Nur77 at time 0.

We agree with your viewpoint that “control of PEPCK expression by Nur77 over time without CHX would be more informative.” To this end, we performed an experiment in which SUMO/Ubc9 and Nur77 were transfected together into cells, and we analyzed PEPCK levels at different times without CHX treatment. The result indicated that endogenous PEPCK1 levels could be elevated by Nur77 at different Nur77 transfection times (Figure 5c). Thus, PEPCK1 expression can be regulated by Nur77.

As you mentioned, the results in Supplementary Figure 5c (original presented in Supplementary Figure 5a) were not from the same blot, so it is not suitable to compare PEPCK1 protein levels with or without Nur77 overexpression at time 0. The effect of Nur77 on PEPCK1 expression without CHX treatment is shown in Figure 5b.

Other issues:

17. Reviewing the manuscript was very difficult since the size of the figures was so small. In addition, the manuscript is need of significant copy editing.

The Figures have been enlarged in the revised manuscript. The manuscript has also been edited by Nature Research Editing Service, and the relative certification has been included as below.

Nature Research Editing Service Certification

This is to certify that the manuscript titled Nuclear receptor Nur77 attenuates PEPCK1 sumoylation to suppress hepatocellular carcinoma via switching glucose metabolism was edited for English language usage, grammar, spelling and punctuation by one or more native English-speaking editors at Nature Research Editing Service. The editors focused on correcting improper language and rephrasing awkward sentences, using their scientific training to point out passages that were confusing or vague. Every effort has been made to ensure that neither the research content nor the authors' intentions were altered in any way during the editing process.

Documents receiving this certification should be English-ready for publication; however, please note that the author has the ability to accept or reject our suggestions and changes. To verify the final edited version, please visit our verification page. If you have any questions or concerns over this edited document, please contact Nature Research Editing Service at support@as.springernature.com.

Manuscript title: Nuclear receptor Nur77 attenuates PEPCK1 sumoylation to suppress hepatocellular carcinoma via switching glucose metabolism

Authors: Xue-li Bian^{1*}, Hang-zi Chen^{1*}, Peng-bo Yang^{1*}, Ying-ping Li^{1*}, Fen-na Zhang, Jia-yuan Zhang¹, Wei-jia Wang¹, Wen-xiu Zhao², Sheng Zhang², Qi-tao Chen¹, Yu Zheng¹, Xiao-yu Sun¹, Xiao-min Wang², Kun-Yi Chien³, Qiao Wu^{1**}

Key: 1FB0-7CA2-B6E5-FCEE-1300

This certificate may be verified at secure.authorservices.springernature.com/certificate/verify.

18. *A number of statements that are made are overly stated*

Eg. "anacardic acid markedly stimulated..." there was a small change- not markedly.

We have revised the overstated descriptions. Thank you for your helpful comment.

19. *It is unclear how the authors can make the statement on line 215. Both ALLN and MG132 will inhibit proteosomal degradation. Therefore, this raises the questions as to how ALLN did not have a similar effect as MG132.*

We apologize for mistakenly labeling "ALLM" as "ALLN." The compounds we used in Figure 3h were MG132 and ALLM. ALLM is a calpain inhibitor, while MG132 is a proteasome inhibitor.

Responses to Reviewer #2

1- Given the impact of PEPCK1 on T-cell function (Ho et al. Cell 2015) and the importance of the chronic inflammatory liver disease for HCC development, pre-neoplastic stages of hepatocarcinogenesis should be explored in the context of the here suggested study. Further, clinic-pathological information for the investigated specimens should be added. Further, the data for the Nur77 expression in the investigated HCC cohort should be shown.

Thank you for your suggestion. To display the pre-neoplastic stages of hepatocarcinogenesis, we detected hepatic cirrhosis in DEN/CCl₄-induced HCC mouse samples by PicroSirius Red staining, which showed that more hepatic fibrosis was detected in Nur77-KO mice than in WT mice (Supplemental Figure 1c). Consistently, Nur77-KO mice also developed more severe inflammatory status in this DEN/CCl₄ model, as indicated by higher circulating IL-6 levels (Supplemental Figure 1d). Because inflammation and cirrhosis are considered major risk factors for hepatocarcinogenesis, our results suggested that Nur77 not only inhibits HCC development but also retards HCC initiation.

The clinic-pathological information and Nur77 immunoreactive scores in the investigated specimens are shown in Supplemental Table 2.

2- Throughout the manuscript the utilized of cell lines are inconsistently applied (e.g. Figure 1 and 2). This should be extended, in particular for the investigation where only hepatoma cell line (and 293T cells) was used (e.g. Figure 4). Further, it would be interesting to include untransformed hepatocytes as a negative control. Furthermore, given the dominant role of p53 (also suggested in the introduction) on gluconeogenesis and the high incidence of p53 mutations in HCC, it might be interesting to explore this issue. This could be achieved, e.g. by adding the p53-null cell line Hep3B. Of note, SK-Hep1 is a cell line of endothelial origin (Heffelfinger et al. 1992) and should not be used synonymously in the context with HCC cell line. The cell line should be replaced.

In this manuscript, we mainly used HepG2 (higher Nur77 expression), SMMC-7721 and Huh7 (lower Nur77 expression) to analyze the biological functions of Nur77 and its effects on PEPCK1. 293T cells were mainly used for transfections due to their higher transfection efficiency. To make the utilized cell lines consistent, we added some critical experiments in hepatoma cell lines in Figure 1f, 3d, 4b, 5a and Supplementary Figure 2b, 2d, 3d, 4a, 4e in the revised version.

According to the suggestion, we also tested the functions of Nur77 in untransformed hepatocyte L02 cells. However, we unexpectedly found that in L02 cells, Nur77 was mainly

in the nucleus (Supplemental Figure 2d). L02 cells displayed a resistance to Nur77-mediated repression of cell proliferation (Supplemental Figure 1k), although Nur77 showed similar regulatory effects on glucose metabolism in L02 cells (Supplemental Figure 2c) as in liver cancer cells (Figure 2a, bottom). In addition, Nur77 overexpression regulated the expression of its target gene E2F1 in L02 cells but not in liver cancer cell lines (Supplementary Figure 2e). These results suggest that the Nur77 regulatory mechanism in L02 cells is different from that in liver cancer cells.

A p53-null cell line Hep3B was used to analyze the p53 role in the current study. The results indicated that Nur77 overexpression still inhibited clone formation and regulated gluconeogenesis (Supplemental Figure 1m & 2b), similar to the effects observed in the HepG2 cell line (WT p53 expression). Therefore, Nur77 exerted its functions independent of p53 in our study.

According to the suggestion, we deleted the SK-Hep1 cell line and added three additional liver cancer cell lines in the revised version of the manuscript (Supplemental Figure 1i).

3- The authors suggest a direct regulatory effect of Nur77 on PEPCK1 (Supplemental Figure 2). While this is certainly interesting, mechanistic investigations confirming the suggested direct regulatory effect are missing. This should be extended to confirm the Nur77-PEPCK1 regulatory role for HCC. Further, despite the potential regulation of PEPCK1 it remains unclear how Nur77 affects the tumorigenic potential of HCC cells, e.g. by regulating WNT signaling (Chen et al. Gut 2012). In line with this, impact of Nur77 on sumoylation and the suggested mechanism of operation not striking and most pronounced in non-hepatoma 293T cells (Figure 5). Quantification and statistical evaluation is needed. Similar, statistical analyses of (e.g. Supplemental Figure 5A) seem necessary, since different base line levels of PEPCK1 expression are shown.

To confirm the Nur77-PEPCK1 regulatory role for HCC, we performed colony-formation assays, which showed that Nur77 and Nur77 2G only inhibited colony formation in controls, but not PEPCK1-knockdown HCC cells (Figure 2i, Supplementary Figure 2p), suggesting a role of PEPCK1 mediation. Furthermore, the xenograft tumor assay in nude mice showed that Nur77 substantially inhibited the PEPCK1-expressing xenograft tumor growth, but slightly influenced the PEPCK1 K124R-expressing tumor growth (Figure 5f), further emphasizing that the Nur77 repression of HCC depends on the sumoylation of PEPCK1 at Lys124.

To determine whether Nur77 also inhibits Wnt signaling to retard HCC growth, the plasmid dnTCF4, which has been shown to inhibit Wnt signaling activity (*Mol Cell Biol.* 1999;19(8):5696-706), was constructed. Nur77 could efficiently inhibit cell proliferation in both wildtype HCC cells and Wnt signaling-inactivated HCC cells that were transfected with dnTCF4 (Supplementary Figure 11), suggesting that Wnt signaling is not required for Nur77 to inhibit HCC in our study.

In this version, we deleted some of the experiments performed in 293T cells and supplemented additional experiments using HCC cell lines, including Figures 3c, 3d, 4b, 5a and 5c.

Necessary quantification and statistical evaluations have been included in Figure 3E, 3G, and Supplementary Figure 3e, 3g, 4d and 5c, according to our three parallel experimental results.

4- Given the plethora of different curcumin targets, the mechanisms of Nur77 upregulation by curcumin are unclear and should be evaluated. Since curcumin is known to inhibit cell growth of hepatoma cells by affecting multiple molecular targets, specificity for the suggested findings should be confirmed by a more targeted approach.

Curcumin has multiple molecular targets to exert its functions, and its functions in cancer are complex. According to the comments from you and Reviewer #1, the Editor has suggested deleting the content of the curcumin study. We agree with this suggestion.

5- Promotor methylation of Nur77 should be confirmed in the cell lines (Figure 6). Rational for Snail selection should be explained in more detail.

We added additional experiments to detect Nur77 promoter methylation in different HCC cell lines (Figure 6b, bottom).

Snail is a suppressive transcription factor that binds to E-box DNA sequences to silence gene expression through the recruitment of chromatin remodelers and DNA methyltransferases (*Nat Rev Mol Cell Biol.* 2014; 15(3):178–196; *The Journal of clinical investigation* 2012, 122(4): 1469-1486; *Cancer cell* 2013, 23(3): 316-331). Snail-mediated epigenetic regulation was recently reported to be involved in hepatocarcinogenesis (*Cell Death Differ.* 2016;23(4):616-27). After sequence analysis, we found at least five Snail-binding E-boxes (CAGGTG) in the proximal region of the Nur77 promoter (Supplementary Figure 6a), indicating that Snail may be involved in the epigenetic regulation of Nur77 in HCC. We have rewritten the Snail-associated paragraph on page 17.

Minor comments

6- The study by Wurmbach et al. includes 4 stages of HCC. Please explain the differences in the here presented study. Please also explain how the statistical analyses shown in the Figure have been derived.

The raw Wurmbach liver data from Oncomine includes 6 different stages of liver tissues: normal liver, cirrhosis, liver cell dysplasia, stage 1 HCC, stage 2 HCC and stage 3 HCC. To study Nur77 expression in normal liver tissue vs. HCC, we applied the data from normal liver, stage 1 HCC, stage 2 HCC and stage 3 HCC and presented them in Supplementary Fig. 1a. Significance was determined with one-way ANOVA followed by Tukey's post hoc test.

7- The effect of different Nur77 levels on proliferative status in individual tumors (e.g. by Ki67 and PCNA) should be confirmed.

To confirm the effect of different Nur77 levels on proliferative status, we analyzed PCNA protein levels in clinical samples. As expected, Nur77 and PCNA protein showed a clear negative correlation (Supplemental Figure 1b).

8- The authors suggest that Nur77 is driven by non-genomic events. Please extend the explanation of this hypothesis. Maybe data supporting that Nur77 is not impaired by genetic alterations (e.g. by using publically available repositories) can be included.

Nur77 not only acts as a transcription factor to positively or negatively regulate downstream gene expressions but also performs its non-genomic actions independent of its regulation of downstream target genes. This functional mode of Nur77 has been demonstrated recently by several reviews (*Steroids*. 2015 Mar; 95:1-6; *Expert Opin Ther Targets*. 2012; 16(6):573-85). In this study, we showed that Nur77 could not influence the transcription level of PEPCK1 (Supplementary Figure 5b), and Nur77 2G, a Nur77 mutant without transcriptional activity, could still inhibit HCC growth (Supplementary Figure 2f & 2p). These results suggested that transcriptional activity is not involved in Nur77 inhibition of HCC. Instead, Nur77 mainly inhibited HCC via interaction with PEPCK1 and stability of PEPCK1 protein through inhibition of its sumoylation. To avoid misunderstanding, we have rewritten the sentences on Page 9.

9- It is unclear how the doses for the used compounds were established in the respective cell lines. Experimental data or suitable references should be provided.

We have tested the effect of different compounds with the concentration according to relevant references. For example, we used the doses of anacardic acid according to *Chemistry & Biology*. 2009, 16:133–140; the doses of CHX according *Oncogene*. 2014, 33:5303–5309; the doses of MG-132 according to *Oncotarget*. 2015, 6:10880-9; the dose of ALLM according to *Endocrinology*. 2007, 148:34–44 and the dose of ADC according to *PLoS ONE* 6: e19862. The concentration of different compounds are now indicated in corresponding figure legends.

10- Number of replicates and statistical analyses should be added wherever necessary.

We have added this information to the figures as necessary. Thank you for your suggestion.

Responses to Reviewer #3

1. Authors showed that sumoylation down-regulated PEPCK1 stability, which would decrease gluconeogenesis and increase tumorigenesis in HCC. If it is true, it should be detected that PEPCK1 is highly-sumoylated in HCC samples compared to in normal liver tissue, which would strongly support the link between PEPCK1 SUMOylation and tumor suppressor function of Nur77. Additionally, how sumoylation affect PEPCK1 stability?

According to the suggestion, we added two *in vivo* sumoylation detection assays in clinical and mouse samples. In clinical para-carcinoma samples, PEPCK1 expression was higher than in carcinoma samples, while its sumoylation could be hardly detected in para-carcinoma samples (Figure 3b). In mouse samples, sumoylation was detected in DEN/CCL₄-induced HCC samples, but not in normal liver tissues (Figure 3a). As Nur77 expression was lower in carcinoma samples but higher in para-carcinoma samples (Figure 1a), and as Nur77 and PEPCK1 showed a positive correlation (Figure 2c), these results strongly support the link between PEPCK1 sumoylation and the tumor suppressor function of Nur77. Moreover, the fact that Nur77 efficiently inhibited xenograft tumor growth when wildtype PEPCK1 was co-expressed, but only moderately inhibited tumor growth when PEPCK1 K124R was co-expressed (Figure 5f), further confirmed the suppressive function of Nur77 on HCC relied on its regulation of PEPCK1 sumoylation.

Our results indicated that sumoylation affects PEPCK1 stability through the ubiquitin-proteasome pathway, as MG132 but not ALLM treatment led to PEPCK1 accumulation in the presence of SUMO1/Ubc9 (Figure 3h, left), which was accompanied by

increased endogenous ubiquitin targeting to PEPCK1 but not to PEPCK1^{K124R} (Figure 3h, middle). For comparison, increased ubiquitination was still detected on PEPCK1^{K471&473R} (Figure 3h, right), which is another indication that Lys124 is the only critical sumoylation site for PEPCK1 stability.

2. It has been shown that sumoylation of PEPCK1 decreases tumorigenesis in HCC. Therefore, sumoylation mutant of PEPCK1 would rescue the tumorigenesis in Nur77-silenced HCC. They should do that to conclusively make statement that Nur 77 suppress HCC via PEPCK1 sumoylation.

To further verify Nur77 repression in HCC via PEPCK1 sumoylation, we performed xenograft experiments in nude mice. As shown in Figure 5f, Nur77 efficiently inhibited PEPCK1-expressed xenograft tumor growth, but only slightly influenced PEPCK1 K124R-expressed tumor growth, further suggesting that Nur77 suppressed HCC via regulation of HCC PEPCK1 sumoylation at Lys124.

3. It is of interesting that p300 acetylation UBC9 increases PEPCK1 sumoylation. Does p300-UBC9 regulation affect global sumoylation? It seems that acetylated UBC9 has more affinity to bind to PEPCK1 than un-acetylated form. They should prove it by binding experiment.

According to your suggestion, we detected global sumoylation in several HCC cells, and the results indicated that transfection of p300 could not affect the global sumoylation induced by Ubc9 (Supplemental Figure 4e). It appears that p300-mediated Ubc9 acetylation selectively influences PEPCK1 sumoylation.

The acetylation of Ubc9 at Lys65 by p300 facilitates its binding with PEPCK1, as revealed in Figure 4e, in which the transfection of p300 enhanced the PEPCK1-Ubc9 interaction, but not the PEPCK1-Ubc9^{K65R} interaction. However, PEPCK1 interacted with both Ubc9 and its mutant Ubc9^{K65R} with similar affinity (Supplemental Figure 4g) without the presence of p300, which further indicates that p300, through acetylating Ubc9 and facilitating acetylated Ubc9 binding to PEPCK1, could enhance PEPCK1 sumoylation.

4. The authors mention that hepatic expression of Nur77 can elevate gluconeogenesis through transcriptional upregulation of gluconeogenic genes, including G6pc, Fbp1 and Fbp2, and Eno317. They also show that mutation of Nur77 binding to DNA affect the growth of SMMC-7721 cells (Supplemental Figure 2B), suggesting that the transcription

activity of Nur77 also engage the tumorigenesis. How to comment the contribution of Nur77-PEPCK1 non-transcription activity and Nur77 transcription activity to HCC?

It has been reported that Nur77 can transcriptionally activate the expression of several gluconeogenic genes in normal liver tissue (*Nature medicine* 2006, 12(9): 1048-1055). However, Nur77 appears to lose most of its transcriptional activity in HCC cells, as revealed by the fact that Nur77 was mainly localized in cytoplasm (Supplementary Figure 2d). Nur77 only slightly enhanced the transcription of Fbp2 and Eno3, but could not influence G6pc and Fbp1 transcription (Supplementary Figure 5a) in several HCC cells. In addition, Nur77 could not induce PEPCK1 gene expression in different HCC cell lines (Supplementary Figure 5b). Combined with the results that Nur77 2G, a transcription inactivated mutant, could also efficiently inhibit HCC cell proliferation via PEPCK1 mediation (Supplementary Figure 2f & 2p), it is likely that although the transcriptional activity of Nur77 may partially involve in HCC suppression, Nur77 mainly exerts its function to suppress HCC via protein-protein interaction, such as Nur77-PEPCK1.

Minor:

4. In Fig 3a, Is there any changes in PEPCK1 sumoylation in liver with or without DEN/CCL4 treatment?

We have added this experiment. In mouse liver samples, sumoylation could be detected in DEN/CCL4-induced HCC, but not in normal liver tissues (Figure 3a).

5. In Fig 4, Does knock-down p300 affect PEPCK1 expression as well as sumoylation in HCC cells?

As shown in Supplementary Figure 4a, knockdown of p300 did affect PEPCK1 expression or sumoylation in transfection or endogenous experiments in different HCC cell lines.

6. In Fig 4a (bottom), overexpression of p300 decreased endogenous PEPCK1 expression in 293T and HepG2, did affect the sumoylation of endogenous PEPCK1?

Overexpression of p300 increased endogenous PEPCK1 sumoylation, accompanied by decreased endogenous PEPCK1 expression, in different HCC cell lines (now presented in Figure 4b).

7. The author mentioned in the discussion that the third mechanism of Nur77 regulating

PEPCK1 is attenuating p300-induced PEPCK1 acetylation, is there any evidence for it?

Please see Figure 5e, which indicates that Nur77 overexpression abolished p300-induced PEPCK1 acetylation.

REVIEWERS' COMMENTS:

Reviewer #1 (Remarks to the Author):

The authors have done an very good job of addressing my concerns.

Reviewer #2 (Remarks to the Author):

This is the revised version of the manuscript by Bian et al..

The authors experimentally addressed several of the key comments and clarified several unclear aspects. The revision significantly improved the manuscript and strengthened the major conclusions.

However, some questions still remain:

The importance of the here proposed mechanisms during sequential hepatocarcinogenesis in humans, in particular in pre-neoplastic stages, is still unclear (i.e. chronic liver disease, cirrhosis, dysplasia,...). Further, quantification of fibrosis (Figure S1c) is needed and magnification should be indicated (this applies to all figures).

The authors convincingly demonstrate that Wnt is not required for Nur77-dependent antitumorigenic potential. However, it still remains unclear what exact mechanisms despite PEPC1 regulation confer to this property. This should be discussed in more detail. Quantification of key results should be added to support the impact of Nur77 on sumoylation.

Reviewer #3 (Remarks to the Author):

I am satisfied with the answers for my comments. No more questions.

REVIEWERS' COMMENTS:

Reviewer #1 (Remarks to the Author):

The authors have done an very good job of addressing my concerns.

Reviewer #2 (Remarks to the Author):

This is the revised version of the manuscript by Bian et al..

The authors experimentally addressed several of the key comments and clarified several unclear aspects. The revision significantly improved the manuscript and strengthened the major conclusions.

However, some questions still remain:

The importance of the here proposed mechanisms during sequential hepatocarcinogenesis in humans, in particular in pre-neoplastic stages, is still unclear (i.e. chronic liver disease, cirrhosis, dysplasia,...). Further, quantification of fibrosis (Figure S1c) is needed and magnification should be indicated (this applies to all figures).

The authors convincingly demonstrate that Wnt is not required for Nur77-dependent antitumorigenic potential. However, it still remains unclear what exact mechanisms despite PEPCCK1 regulation confer to this property. This should be discussed in more detail. Quantification of key results should be added to support the impact of Nur77 on sumoylation.

Reviewer #3 (Remarks to the Author):

I am satisfied with the answers for my comments. No more questions.

Response to Reviewer #2:

The importance of the here proposed mechanisms during sequential hepatocarcinogenesis in humans, in particular in pre-neoplastic stages, is still unclear (i.e. chronic liver disease, cirrhosis, dysplasia,...). Further, quantification of fibrosis (Figure S1c) is needed and magnification should be indicated (this applies to all figures).

The authors convincingly demonstrate that Wnt is not required for Nur77-dependent antitumorigenic potential. However, it still remains unclear what exact mechanisms despite PEPCK1 regulation confer to this property. This should be discussed in more detail. Quantification of key results should be added to support the impact of Nur77 on sumoylation.

Thank you for your suggestion!

1) As we know, development from pre-neoplastic stages (i.e. chronic liver disease, cirrhosis, dysplasia...) to cancer is a long and complex process. Different stages have different features that required a lot of experiments to demonstrate. Although, in the first round of revision, four panels (Supplementary Figure 1C, D, E, and H) are not enough to clarify the mechanisms of sequential hepatocarcinogenesis, these data at least indicated that Nur77 may participate in the regulation of NAFLD and cirrhosis. Since this current study mainly focuses on hepatocarcinogenesis rather than pre-neoplastic stages, we suggested not to paid more attention on pre-neoplastic stages. Here, we supply some Oncomine data, which indicates that Nur77 expression remained comparable between normal liver tissue and cirrhosis liver tissue, but was downregulated in tissues of liver cell dysplasia (Supplementary Figure 1a, right).

2) Quantification of fibrosis (Figure S1c) is added and the scale bars are indicated to all figures necessary (Figure 1a, 1d, 1e, 2b, 2f, 2g, 5f, 6c and Supplementary Figure 1c, 1e, 1f, 1g).

3) As for the question “what exact mechanisms despite PEPCK1 regulation confer to this property”, we have added more description in DISCUSSION (page 19-20).

4) Quantification of key results to support the impact of Nur77 on sumoylation are indicated in Figure 5A.